# Nucleotidyltransferase toxin MenT extends aminoacyl acceptor ends of serine tRNAs to control *Mycobacterium tuberculosis* growth

Xibing Xu [1] ✉, Roland Barriot [1], Bertille Voisin [2], Tom J. Arrowsmith[3], Ben Usher[3], Claude Gutierrez [2], Xue Han[1], Carine Pagès[1], Peter Redder [1], Tim R. Blower [3], Olivier Neyrolles [2] & Pierre Genevaux [1] ✉

Toxins of toxin-antitoxin systems use diverse mechanisms to inhibit bacterial growth. In this study, we characterize the translation inhibitor toxin MenT3 of *Mycobacterium tuberculosis*, the bacterium responsible for tuberculosis in humans. We show that MenT3 is a robust cytidine specific tRNA nucleotidyltransferase in vitro, capable of modifying the aminoacyl acceptor ends of most tRNA but with a marked preference for tRNA$^{Ser}$, to which long stretches of cytidines are added. Furthermore, transcriptomic-wide analysis of MenT3 targets in *M. tuberculosis* identifies tRNA$^{Ser}$ as the sole target of MenT3 and reveals significant detoxification attempts by the essential CCA-adding enzyme PcnA in response to MenT3. Finally, under physiological conditions, only in the presence the native *menAT3* operon, an active pool of endogenous MenT3 targeting tRNA$^{Ser}$ in *M. tuberculosis* is detected, likely reflecting the importance of MenT3 during infection.

Toxin-antitoxin (TA) systems are stress-responsive genetic elements encoding for a deleterious toxin and its antagonistic antitoxin. They are widespread in bacterial genomes and mobile genetic elements[1–3]. They have roles in defending bacteria against phage infection and in the maintenance of genomic regions and plasmids, and in some cases, they have been shown to contribute to bacterial virulence and antibiotic persistence[4–9]. Under steady-state conditions, toxin activity is blocked by its antagonistic antitoxin, and bacterial growth is not detectably impacted. Yet, under specific conditions, including phage infection or the loss of plasmids, the TA equilibrium can be significantly unbalanced in favor of the more stable toxin. As a consequence, the free active toxin can target essential cellular processes or structures, mainly translation, DNA replication, metabolism, or the cell envelope, causing growth inhibition or cell death[2,3].

Tuberculosis is the second cause of death due to an infectious agent, after COVID-19. According to the most recent WHO report (2023), over 10 million people fell ill with tuberculosis in 2022, and 1.3 million died from the disease. The increasing occurrence of multi and extensively drug-resistant strains of the causative *Mycobacterium tuberculosis* bacterium has greatly heightened the need for the development of new drugs and new treatment strategies. *M. tuberculosis* encodes an unusually high number of TA systems, over 86, representing close to 4% of its genome[10,11]. This includes multiple homologs from conserved TA families that have been shown to be induced under relevant stress conditions, including hypoxia, macrophage engulfment, or drug exposure[12–14]. Although their contribution to *M. tuberculosis* physiology and virulence is currently unknown, it has been proposed that activated toxins could modulate *M. tuberculosis* growth under certain conditions, thereby contributing to survival in the human host[11,13,15]. The deleterious nature of many *M. tuberculosis* toxins has raised the possibility that new antibacterial properties demonstrated by toxins might be exploited to identify new drug targets or applied directly as antimicrobials, alone or in combination with standard antibacterial therapy[16–18].

*M. tuberculosis* encodes four MenAT TA systems, comprised of a nucleotidyltransferase (NTase) toxin and a cognate antitoxin

[1]Laboratoire de Microbiologie et Génétique Moléculaires (LMGM), Centre de Biologie Intégrative (CBI), Université de Toulouse, CNRS, UPS, Toulouse, France. [2]Institut de Pharmacologie et de Biologie Structurale (IPBS), Université de Toulouse, CNRS, UPS, Toulouse, France. [3]Department of Biosciences, Durham University, South Road, Durham DH1 3LE, UK. ✉e-mail: xibing.xu@univ-tlse3.fr; pierre.genevaux@univ-tlse3.fr

belonging to one of three different families[19–21]. Although, *menAT2* was recently shown to be required for *M. tuberculosis* pathogenesis in guinea pigs[22], only *menAT1* and *menAT3* were shown to act as bona fide TA systems in their native host *M. tuberculosis*[20,21]. The MenA3 antitoxin inhibits MenT3 through phosphorylation of a serine residue in the catalytic site[23], whilst MenA1 forms an asymmetric heterotrimeric complex with two MenT1 protomers, suggesting a different mode of inhibition[20]. In vitro, the MenT1, MenT3, and MenT4 toxins were shown to inhibit translation by acting as tRNA NTases[20,21]. Yet, biochemical characterization revealed significant differences in MenT toxin specificity for tRNA targets and nucleotide substrates. Though both MenT1 and MenT3 inhibit aminoacylation by transferring pyrimidines (preferentially cytidines) to the 3′-CCA acceptor stems of tRNAs, MenT3 displays a strong preference for serine tRNAs and MenT1 showed no apparent preference[20,21]. MenT4 also modifies the 3′-CCA motif of tRNA acceptor stems but with a preference for GTP as substrate[20]. We have previously demonstrated that MenT3 is by far the most toxic of the four MenT toxins of *M. tuberculosis*, inducing a rapid and efficient self-poisoning[21]. Despite the accumulated knowledge concerning MenT3 in vitro activity and structure, nothing is currently known about MenT3 activity and cellular targets in *M. tuberculosis*.

In this work, we investigate the impact of MenT3 on *M. tuberculosis*, focusing on the identification of tRNA targets on a transcriptomic scale, on the molecular determinants necessary for tRNA recognition, and on the mechanisms by which *M. tuberculosis* counteracts noxious tRNA modification. We show that MenT3 is an effective NTase capable of efficiently modifying all the tRNA in vitro, with a strong preference for tRNA$^{Ser}$. Remarkably, when expressed in *M. tuberculosis*, MenT3 specifically and efficiently adds cytidines to all tRNA$^{Ser}$, without modification detected for the other tRNAs. In sharp contrast with in vitro, different tRNA$^{Ser}$ 3′-end species accumulate in *M. tuberculosis* due to the presence of a CCA-adding enzyme, which appears to be a key factor in the cellular response to MenT3 activity. Importantly, we also identify a steady state level of cytidine addition of tRNA$^{Ser}$ in *M. tuberculosis* wild type under standard growth conditions, and thus solely in the presence of the native chromosomal *menAT3* operon copy. The role of such an unexpected basal level of MenT3-dependent tRNA$^{Ser}$ 3′-end elongation and its possible link with *M. tuberculosis* physiology and virulence is discussed.

## Results

### MenT3 is a promiscuous tRNA NTase with robust poly-C activity in vitro

We previously showed that MenT3 can add two to three cytidines or uridines to in vitro transcribed tRNAs[21]. In addition, individual screening of each of the 45 in vitro purified tRNA of *M. tuberculosis* showed that MenT3 could modify the four tRNA$^{Ser}$ and to a lesser extent tRNA$^{Leu-5}$, while all the other tRNAs were not detectably modified. Yet, our recent observation that mature tRNA were more efficiently modified in vitro by the related MenT1 toxin[20] led us to reinvestigate the nucleotide and tRNA specificities of MenT3. Incubation of *M. smegmatis* tRNA extract with MenT3 in the presence of each individual labeled nucleotide indicates that although UTP (and to a much lesser extent ATP) can be used by MenT3, CTP is by far the preferred nucleotide substrate of MenT3 in vitro (Fig. 1a). In addition, we found that MenT3 is capable of efficiently modifying *M. tuberculosis*, *E. coli* and human tRNAs purified from cell extracts, indicating that MenT3 is a promiscuous tRNA NTase in vitro (Fig. 1b). Furthermore, the NTase activity of MenT3 toward tRNA extracts in vitro is significantly more robust than those of MenT1 or MenT4 (Fig. 1c), which is in agreement with the strong toxicity of MenT3, when compared to the other MenT toxins tested so far[20,21].

We next applied 3′-OH tRNA-seq to the *M. smegmatis* RNA extracts following incubation in the presence or absence of MenT3 and found that over 95% of the pool of detected tRNA was modified with an

average of two to five cytidines added to 3′CCA tRNA ends (CCA + C*n*; Fig. 1d). Remarkably, while most of the tRNAs had these two to five cytidine extensions, the detected tRNA$^{Ser-2}$ and tRNA$^{Ser-3}$ showed a completely different behavior, with significantly longer poly-cytidine (poly-C) extensions of up to $n = 12$ under such conditions (Fig. 1e, Source Data file). Note that tRNA$^{Ser-1}$ and tRNA$^{Ser-4}$ could not be sufficiently detected in *M. smegmatis* extracts. In addition, the *M. smegmatis* selenocysteine tRNA$^{Sec}$, which is not present in *M. tuberculosis*[24] and is very similar to tRNA$^{Ser}$ was also highly modified by MenT3. Together, these in vitro data show that MenT3 is a very robust CTP-dependent tRNA NTase toxin, which can modify most tRNAs with 3′ CCA + C$_{(2-5)}$ extensions, but with a preference for tRNA$^{Ser}$ (or tRNA$^{Sec}$ in *M. smegmatis*) with respect to the length of the cytidine extension.

### Serine tRNA variable loop is critical for poly-C extensions in vitro

We further investigated the marked difference in cytidine extension observed between tRNA$^{Ser}$ and other tRNAs in vitro, using purified *M. tuberculosis* tRNA$^{Ser-4}$ and tRNA$^{Met-2}$ as representative tRNAs. In this case, the tRNAs were independently labeled with [α-32P]-CTP and purified using a ribozyme-based cleavage method that generated more homogeneous 3′-OH ends, to avoid the higher heterogeneity generated in transcripts made with T7 RNA polymerase transcription[20]. Labeled tRNAs were individually incubated with MenT3 in the presence of CTP, and samples were analyzed at different time points (Fig. 2a). The data show the addition of cytidines in both cases, with a more rapid accumulation of longer species of modified tRNA$^{Ser-4}$ when compared to tRNA$^{Met-2}$. Analysis of tRNA$^{His}$ and tRNA$^{Leu-3}$ confirmed the accumulation of shorter extensions as observed for tRNA$^{Met-2}$ (Supplementary Fig. S1). Furthermore, 3′-OH tRNA-seq analysis of the 10 min incubation samples of tRNA$^{Ser-4}$ and tRNA$^{Met-2}$ revealed that the majority of the tRNA$^{Ser-4}$ have indeed acquired 11 to 12 cytidines (with a maximum $n = 17$), while tRNA$^{Met-2}$ only two to five (with a majority at $n = 3$) (Fig. 2b). Together these in vitro data demonstrate that although MenT3 is capable of modifying most tRNA in vitro, it shows a strong preference for tRNA$^{Ser}$.

We next investigated the sequence and structural elements within a tRNA that are required for cytidine addition by MenT3 in vitro. Several variants of tRNA$^{Ser-4}$ with sequential deletion or mutation of the 3′-end nucleotides were labeled, purified using the ribozyme-based cleavage method, and incubated with MenT3 in the presence of CTP. Under such conditions, we found that the adenosine amino acceptor of the CCA motif could be deleted or mutated to U, C, or G without detectably affecting MenT3 activity (Fig. 2c and Supplementary Fig. S1). In sharp contrast, both 3′ΔCA and 3′ΔCCA end deletions within tRNA$^{Ser-4}$ fully prevented elongation by MenT3 (Fig. 2c). The 3′-OH tRNA-seq data confirmed that 3′ΔA, but not 3′ΔCA or 3′ΔCCA tRNA$^{Ser-4}$, can be modified by MenT3 (Fig. 2d). Yet, we noticed that cytidine extensions on 3′ΔA were on average slightly shorter than for the wild type 3′CCA tRNA$^{Ser-4}$ (i.e., 9 to 10 cytidines compared 11 to 12), suggesting that the last nucleotide could still have some weak impact on MenT3 activity (Fig. 2d). Together these data demonstrate that 3′CCA and CCΔA ends are bona fide targets for MenT3.

The main structural feature that differentiates serine tRNAs from the other tRNAs in *M. tuberculosis* is the presence of a longer variable loop[21], which we believe could play a role in MenT3 specificity. To answer this, we first deleted the variable loop in tRNA$^{Ser-4}$ and tested the resulting construct in vitro in the presence of MenT3. Remarkably, deletion of the variable loop of tRNA$^{Ser-4}$ abolished the formation of long poly-C extensions, leaving the deleted form of the tRNA$^{Ser-4}$ to be extended as per non-serine tRNAs of *M. tuberculosis* (Fig. 2e). Note that in contrast with tRNA$^{Ser-4}$, the ΔVL tRNA$^{Ser-4}$ variant deleted of the variable loop was not detectably modified by MenT3 when lower MenT3 concentrations were used (Supplementary Fig. S2a). We next asked whether grafting the variable loop of tRNA$^{Ser-4}$ to an unrelated

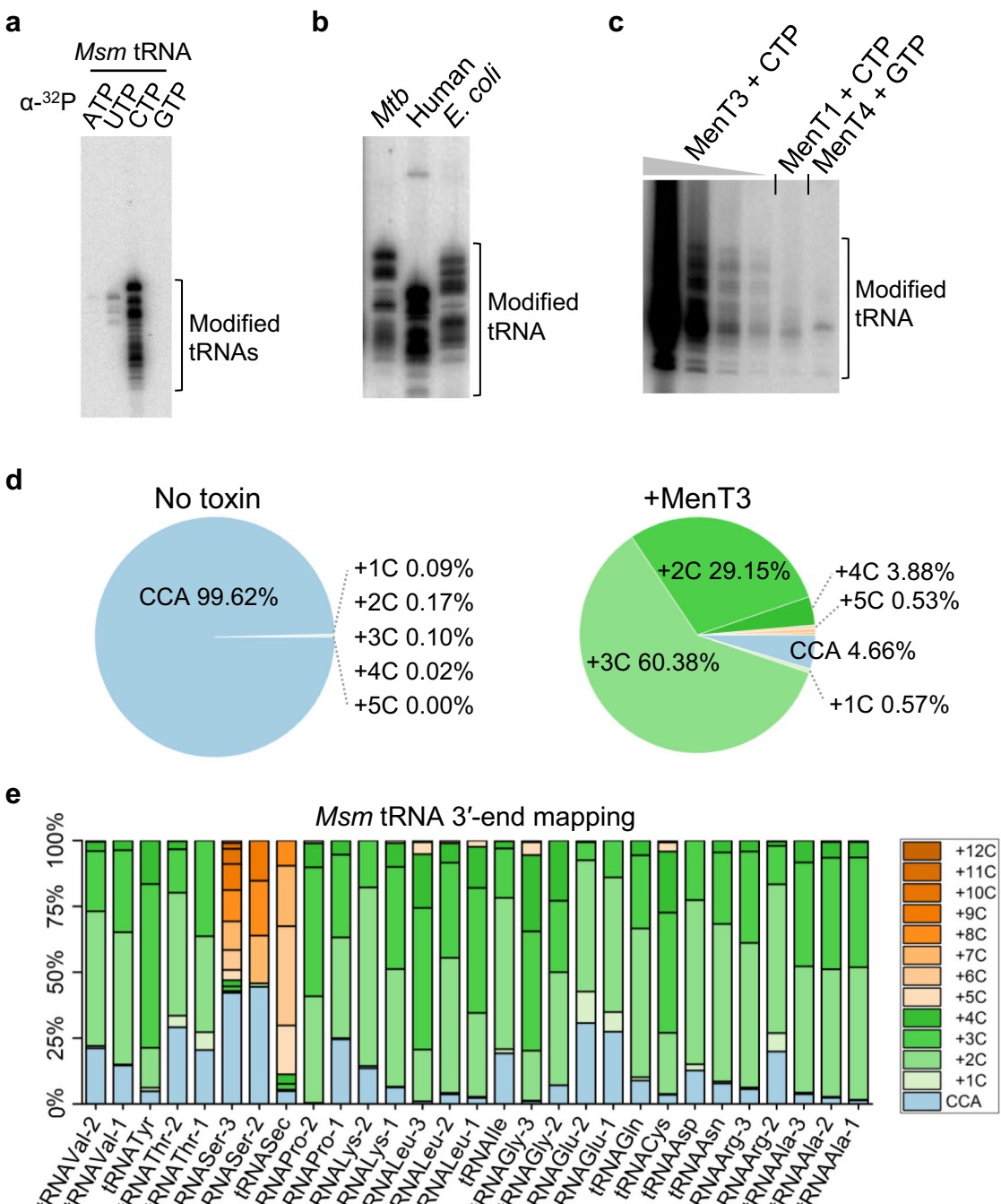

**Fig. 1 | MenT3 NTase activity in vitro. a** MenT3 preferentially adds CTP to tRNA. Total tRNA (100 ng) of *M. smegmatis* (*Msm*) were incubated at 37 °C for 5 min with MenT3 (0.2 μM) and α-32P labeled nucleoside triphosphates (NTPs). **b** Promiscuous tRNA alteration by MenT3. About 1 μg of total RNA from *M. tuberculosis* (*Mtb*) or human cells, or 100 ng of tRNA extracts from *E. coli* were incubated with 0.2 μM MenT3 in the presence of α-32P labeled CTPs for 5 min at 37 °C. **c** Comparison of tRNA NTase activity among MenT toxins. MenT3 (0.2 μM) or MenT1 (5 μM), or MenT4 (5 μM) were incubated with 1 μg of total RNA of *Msm* for 20 min at 37 °C. Given the robust activity of MenT3, the sample was serial diluted 1/10, 1/100, or 1/1000 to facilitate visualization of MenT1 and MenT4 activity through phosphor exposure. Reactions were conducted in the presence of α-32P labeled CTP for

MenT1 and MenT3, and α-32P labeled GTP for MenT4. Representative results of triplicate experiments are shown in panels (**a**–**c**). **d**, **e** tRNA 3′-end mapping. Five μg total RNA from *Msm* were incubated with MenT3 (2.5 μM) and 1 mM CTP for 20 min at 37 °C, and the tRNA-seq library was prepared and sequenced. Modified tRNA reads were quantified in samples with or without MenT3 and the modifications are given as a percentage of the total tRNA 3′-ends. Note that tRNA^Sec is not present in *M. tuberculosis*. The tRNA detected from two independent experiments are shown on the bottom of panel (**e**) and the number of cytidines added is shown in the inset at the right of the same panel (duplicate is shown in Source Data file). Source data are provided as a Source Data file.

tRNA would induce the formation of long poly-C extensions by MenT3. We constructed a tRNA^Met-2 chimera in which the native short variable loop was replaced by the extended variable loop of tRNA^Ser-4, and tested for extension in our in vitro assay. The data presented in Fig. 2e indeed show that the engineered tRNA^Met-2 chimera with the tRNA^Ser-4

variable loop is efficiently modified by MenT3 and accumulates long poly-C extensions, in a manner comparable to that of tRNA^Ser-4 wild type. Analysis by 3′-OH tRNA seq confirmed that the tRNA^Met-2 chimera contained long poly-C extensions (up to *n* = 23) with a majority of 16 to 17 cytidines (Fig. 2f). Together, these data suggest that the variable

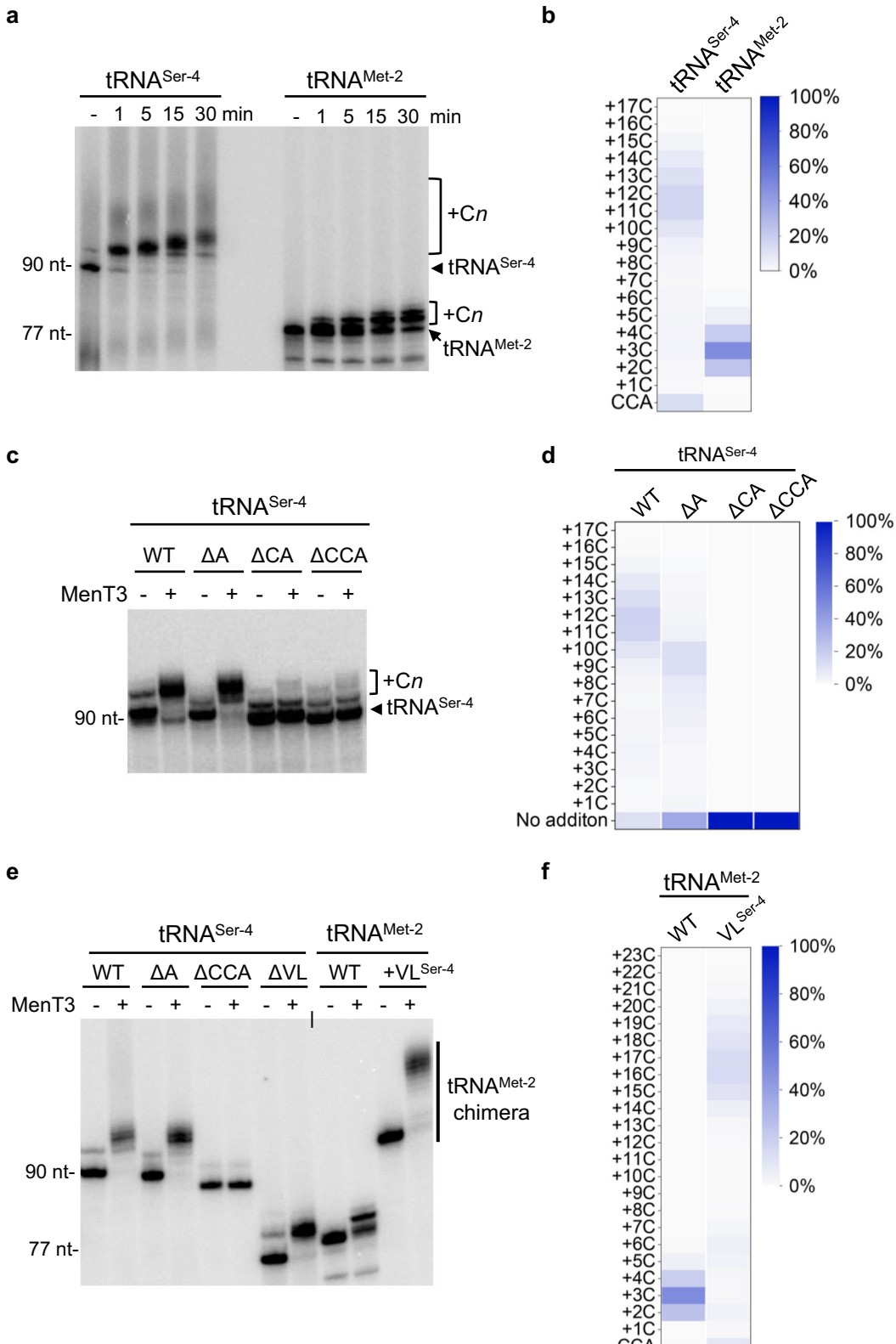

loop is critical for tRNA$^{Ser}$ 3′-end CMP addition and elongation by MenT3. Accordingly, a recent study showed the variable loop of tRNA$^{Ser}$ might indeed act as an anchor that would allow the proper positioning of the CCA 3′-end into the active site of MenT3, thus facilitating efficient CMP incorporation[25].

Some tRNA in *M. tuberculosis* possesses a long variable loop that is comparable to the one of tRNA$^{Ser}$, especially tRNA$^{Leu-2,3 \text{ and } 5}$. Yet, these tRNA are not preferred targets of MenT3 (Fig. 1e[21]. Nevertheless, a comparison of the AlphaFold3 models of tRNA$^{Leu-3}$ and tRNA$^{Ser-4}$ in complex with MenT3 shows that the long variable loop region of both tRNA potentially interacts with the N-terminal region of MenT3 (Fig. 3a), as recently proposed[25]. However, a significantly more reliable interaction network is predicted in the case of tRNA$^{Ser-4}$ (Supplementary Fig. S2b). In order to further investigate what determines such a

**Fig. 2 | tRNA structural and sequence determinants for MenT3 activity in vitro.**
**a** MenT3-tRNA preference in vitro. Purified α-32P labeled tRNA$^{Ser-4}$ and tRNA$^{Met-2}$ were incubated in the presence of 1 mM CTP with MenT3 (0.2 μM) at 37 °C for different time points, separated on a 10% urea gel and revealed by autoradiography. **b** tRNA-Seq mapping of purified tRNA$^{Ser-4}$ and tRNA$^{Met-2}$. Purified tRNA$^{Ser-4}$ and tRNA$^{Met-2}$ (20 ng μl$^{-1}$) were incubated with MenT3 (1 μM) and 1 mM CTP for 10 min at 37 °C and subjected to tRNA-seq. **c** Effect of MenT3 on tRNA$^{Ser-4}$ 3′-end length variants. Purified α-32P labeled 3′ΔCCA, ΔCA, and ΔA truncated tRNA$^{Ser-4}$ were incubated with MenT3 (0.2 μM) for 5 min at 37 °C, separated as in (**a**). **d** tRNA-seq mapping of truncated tRNA$^{Ser-4}$ ΔCCA, ΔCA, and ΔA variants from panel (**b**).

**e** Impact of tRNA$^{Ser}$ variable loop on MenT3 activity. Purified labeled tRNA$^{Ser-4}$ deleted for its long variable loop (ΔVL), and the tRNA$^{Met-2 (VLSer-4)}$ chimera with the long variable loop of tRNA$^{Ser-4}$ (+VL$^{Ser-4}$) were incubated with MenT3 (0.2 μM) for 5 min at 37 °C, separated as in (**a**). **f** tRNA-Seq mapping of purified tRNA$^{Met-2}$ and tRNA$^{Met-2(VLSer-4)}$ chimera from (**e**). Purified tRNA (20 ng μl$^{-1}$) were incubated with MenT3 (1 μM) and 1 mM CTP for 10 min at 37 °C and subjected to tRNA-seq. Representative results of triplicate experiments are shown in (**a**, **c**, **e**), and sequencing data from two independent experiments are shown in panels in (**b**, **d**, **f**). Source data are provided as a Source Data file.

preference for tRNA$^{Ser}$, we first constructed a tRNA$^{Ser-4}$ chimera containing the loop region of the tRNA$^{Leu-3}$ variable loop (Mut I; Fig. 3b) and tested it in vitro in the presence of MenT3. Note that low concentrations of the toxin were used in order to better visualize modifications by MenT3 following migration on urea-PAGE gel, as performed in Supplementary Fig. S2a. The results clearly show that the loop region of the variable loop is not responsible for the preference for tRNA$^{Ser}$ over tRNA$^{Leu}$ (Fig. 3c). We next used this chimera as a template to sequentially delete G/C pairs within the arm of the tRNA$^{Ser-4}$ variable loop (Mut II, III, IV; Fig. 3b) and tested these constructs in vitro in the presence of MenT3. The Mut II construct lacking the first G/C pair, which is equivalent in length to tRNA$^{Leu-3}$, was still efficiently modified by MenT3, indicating that the difference in length between tRNA$^{Ser}$ and tRNA$^{Leu}$ is also not responsible for MenT3 preference under our experimental conditions. Noticeably, while the deletion of two G/C pairs had no significant effect (Mut III), the deletion of three G/C pairs within the arm of tRNA$^{Ser-4}$ significantly impaired cytidine addition by MenT3 (Mut IV; Fig. 3c and Supplementary Fig. S2c), indicating that although the length of the arm is important, it is not the only requirement for MenT3 preference. A close-up view of the proposed interaction between the tRNA$^{Leu-3}$ and tRNA$^{Ser-4}$ in complex with MenT3 shows that the arm of the variable loop of tRNA$^{Leu-3}$ is slightly shifted away from the N-terminal region of MenT3 (Supplementary Fig. S2d), suggesting that the orientation of the variable loop could play a role in MenT3 preference. Remarkably, tRNA$^{Leu-3}$ contains two extra nucleotides G60U61 between the T-arm and the variable loop when compared to tRNA$^{Ser-4}$ (Fig. 3d) and the AlphaFold3 model of tRNA$^{Leu-3}$ deleted of nucleotides G60U61 shows that the arm of the variable loop is now shifted towards the position of tRNA$^{Ser-4}$ arm (Supplementary Fig. S2d). Strikingly, we found that tRNA$^{Leu-3}$ deleted of G60U61 is fully modified by MenT3 in a similar manner as tRNA$^{Ser-4}$ and conversely, insertion of GU nucleotides between the T-arm and the variable loop (at position 62 and 63) of tRNA$^{Ser-4}$ significantly affect modification by MenT3 (Fig. 3c). Together these data reveal the importance of the length and the orientation of variable loop of tRNA$^{Ser}$ for MenT3 activity.

### tRNA$^{Ser}$ is the target of MenT3 in *M. tuberculosis*
We next investigated the cellular tRNA targets of MenT3 in its native host *M. tuberculosis*. MenT3 was expressed in *M. tuberculosis* H37Rv Δ*menAT3* mutant strain for 0, 3, and 24 h at 37 °C, total RNAs were extracted, and 3′-OH tRNA-seq was performed and compared to *M. tuberculosis* H37Rv wild type (Fig. 4a). Using this recently developed method[20], we were able to identify all of the 45 tRNAs of *M. tuberculosis* within the extracts (Fig. 4b and Source Data file). Strikingly, the analysis shows that MenT3 only targets tRNA$^{Ser}$ isoacceptors in *M. tuberculosis*. Modification of tRNA$^{Ser}$ by 3′ addition of cytidines was very robust (approximately 65% of tRNA$^{Ser-2,3,4}$ and 20% of tRNA$^{Ser-1}$ as judged from three independent replicates) after 24 h of expression in *M. tuberculosis* (Supplementary Fig. S3). None of the other tRNAs were detectably modified by MenT3. Similar results were observed in *M. smegmatis* (Supplementary Fig. S4a). These data demonstrate that serine tRNAs are the targets of MenT3 in *M. tuberculosis*.

Unexpectedly, analysis of the seq data revealed that different tRNA$^{Ser}$ 3′-end species were accumulating in *M. tuberculosis* than in vitro (Fig. 4c). First, we found that cytidine extensions were significantly shorter, with the majority of tRNA$^{Ser}$ having only a single added cytidine and a small fraction reaching a maximum of $n = 5$. In addition, while cytidine extensions occur after the adenosine nucleotide of the 3′CCA end (CCA + C$_n$) in vitro, the vast majority, if not all, of the 3′-ends detected were deleted for A or CA, leading mainly to the accumulation of 3′CCΔA + C$_{(1-5)}$ extensions but also to 3′CΔCA without added cytidine (on average 7% tRNA$^{Ser-4}$, 27% tRNA$^{Ser-3}$, 10% tRNA$^{Ser-2}$, and 4% tRNA$^{Ser-1}$ as judged from three independent replicates) after 24 h of expression, which corresponds to only one doubling time for *M. tuberculosis*. Note that the appearance of such species of tRNA$^{Ser}$ was confirmed by a northern blot using a probe against tRNA$^{Ser-3}$ in *M. smegmatis* (Supplementary Fig. S4b). In addition, although not present in *M. tuberculosis*, the data confirm that tRNA$^{Sec}$ can also be targeted by MenT3 in *M. smegmatis* (Supplementary Fig. S4a and Fig. 1e). Remarkably, the fact that 3′CΔCA ends are not modified by MenT3 in vitro (Fig. 2c) strongly suggests that such tRNA species could accumulate in *M. tuberculosis* following the processing of MenT3-modified tRNAs by endogenous tRNA quality control or repair enzymes affecting CCA+C$n$ elongation of tRNA$^{Ser}$ by MenT3[26]. Accordingly, such enzymes could also be responsible for generating 3′CCΔA ends that are efficiently processed by MenT3 in vitro (Fig. 2c) and which accumulate as the major 3′CCΔA + C$_{(1-5)}$ species in *M. tuberculosis* (Fig. 4c).

### CCA-adding enzyme counteracts MenT3-mediated elongation
Previous work performed in *E. coli* showed that co-overexpression of the ribonuclease RpH involved in tRNA 3′-end maturation[26] confers resistance to MenT3[21]. In order to investigate the possible role of *M. tuberculosis* RpH in generating 3′ CCΔA or CΔCA templates in response to MenT3, RpH was purified and incubated with labeled tRNA$^{Ser-4}$ previously modified (or not) by MenT3 and in which CTP has been removed to prevent possible residual NTase activity of MenT3 (Supplementary Fig. S5). Under such conditions, we found that RpH was indeed able to trim the poly-C modified tRNA$^{Ser-4}$ to generate 3′ CCA, CCΔA and CΔCA ends in vitro, suggesting that RpH could potentially be involved in tRNA quality control in response to MenT3. Yet, in sharp contrast with *E. coli*, overexpression of RpH from *M. smegmatis* failed to suppress MenT3 toxicity in *M. smegmatis* (Fig. 5a). In addition, although RpH overexpression in the presence of MenT3 could generate shorter tRNA$^{Ser}$ (CCΔA+C$_n$) ends, it did not lead to a detectable accumulation of 3′ CCA, CCΔA, or CΔCA species (Fig. 5b). This suggests that although RpH is capable of targeting MenT3-modified tRNA$^{Ser}$, it is not the main factor responsible for the accumulation of the truncated tRNA species (Fig. 4).

Only a few tRNA genes of *M. smegmatis* and *M. tuberculosis* encode a CCA triplet at their 3′ termini, i.e., 14 out of 46 and 13 out of 45, respectively[27]. This is in sharp contrast with *E. coli*, in which all the tRNA genes possess such triplet[28]. Accordingly, tRNAs maturation in *M. tuberculosis* required the essential CCA-adding enzyme PcnA[27,29], while it is not the case in *E. coli*[30]. In addition to tRNA maturation, CCA-adding enzymes also contribute to tRNA quality control by targeting

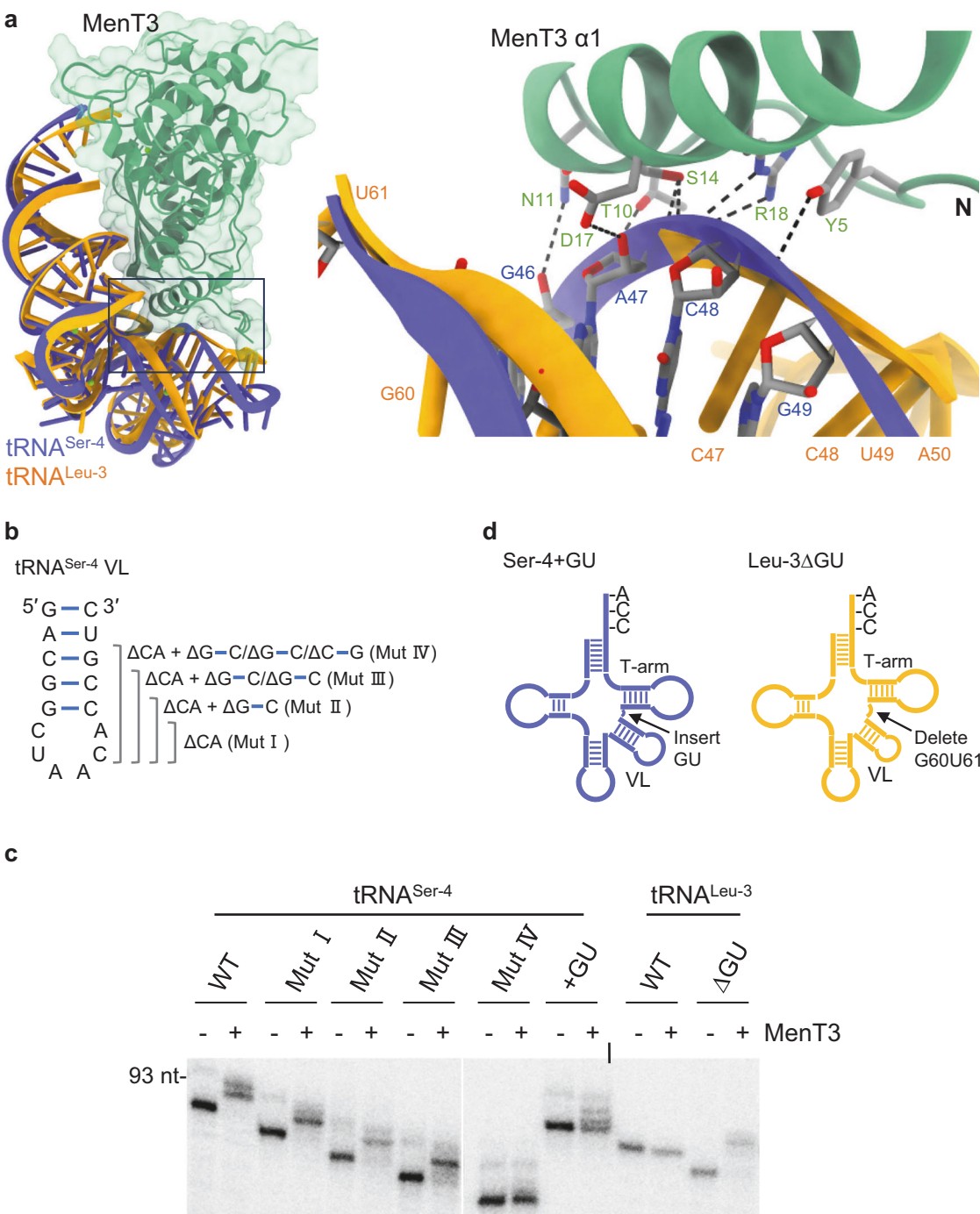

**Fig. 3 | The length and orientation of the tRNA^Ser variable loop contribute to MenT3 preference. a** Model of the interaction between MenT3 and tRNA^Ser-4, or tRNA^Leu-3 generated using AlphaFold3. The pTM/ipTM of MenT3-tRNA^Ser-4 and MenT3-tRNA^Leu-3 were 0.91/0.85 and 0.89/0.77, respectively. A close-up view of the possible interaction between MenT3 and the variable loop of tRNA^Ser-4 and tRNA^Leu-3 is shown on the right panel. Residues in the arm structure of the tRNA^Ser-4 variable loop are depicted as sticks. Potential hydrogen bonds are indicated with black dashed lines, highlighting key interactions. **b** The four deletion mutants of tRNA^Ser-4 used to assess the importance of the length of the variable loop for MenT3-mediated alterations are presented. **c** Effect of MenT3 on tRNA^Ser-4, tRNA^Leu-3 wild-type and mutants. Purified α-32P labeled tRNAs were incubated with MenT3 (0.02 μM) in the presence of 1 mM CTP at 37 °C for 2 min. The modified tRNAs were separated on a 10% urea gel and visualized by autoradiography. A representative gel of three independent experiments is shown. **d** Insertion or deletion between the T-arm and the VL or tRNA^Ser-4 and tRNA^Leu-3, respectively. The tRNA^Ser-4 + GU has a GU dinucleotide insertion between the VL and the T-arm to mimic tRNA^Leu-3, and the tRNA^Leu-3ΔGU has a deletion of GU between the VL and T-arm to mimic tRNA^Ser-4. The effect of MenT3 on the different mutants is shown (**c**). Source data are provided as a Source Data file.

tRNA for degradation[31], or during CCA synthesis, by excision of the 3′ terminal nucleotide in the presence of inorganic pyrophosphate (PPi) in vitro[32,33]. Strikingly, we found that overexpression of PcnA in *M. smegmatis* confers resistance to MenT3 (Fig. 5a). As a control, the mutation in the NTase activity center of PcnA abolished such

suppressive effect, indicating that active PcnA enzyme is required for growth rescue (Supplementary Fig. S4c). Northern blot analysis of tRNA^Ser-3 following co-expression of PcnA and MenT3 confirmed such effect, as judged by the appearance of predominant tRNA^Ser-3 wild-type forms and the disappearance of the C*n* elongations induced by MenT3

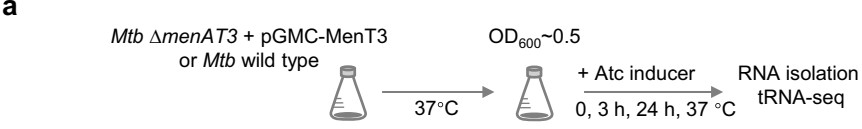

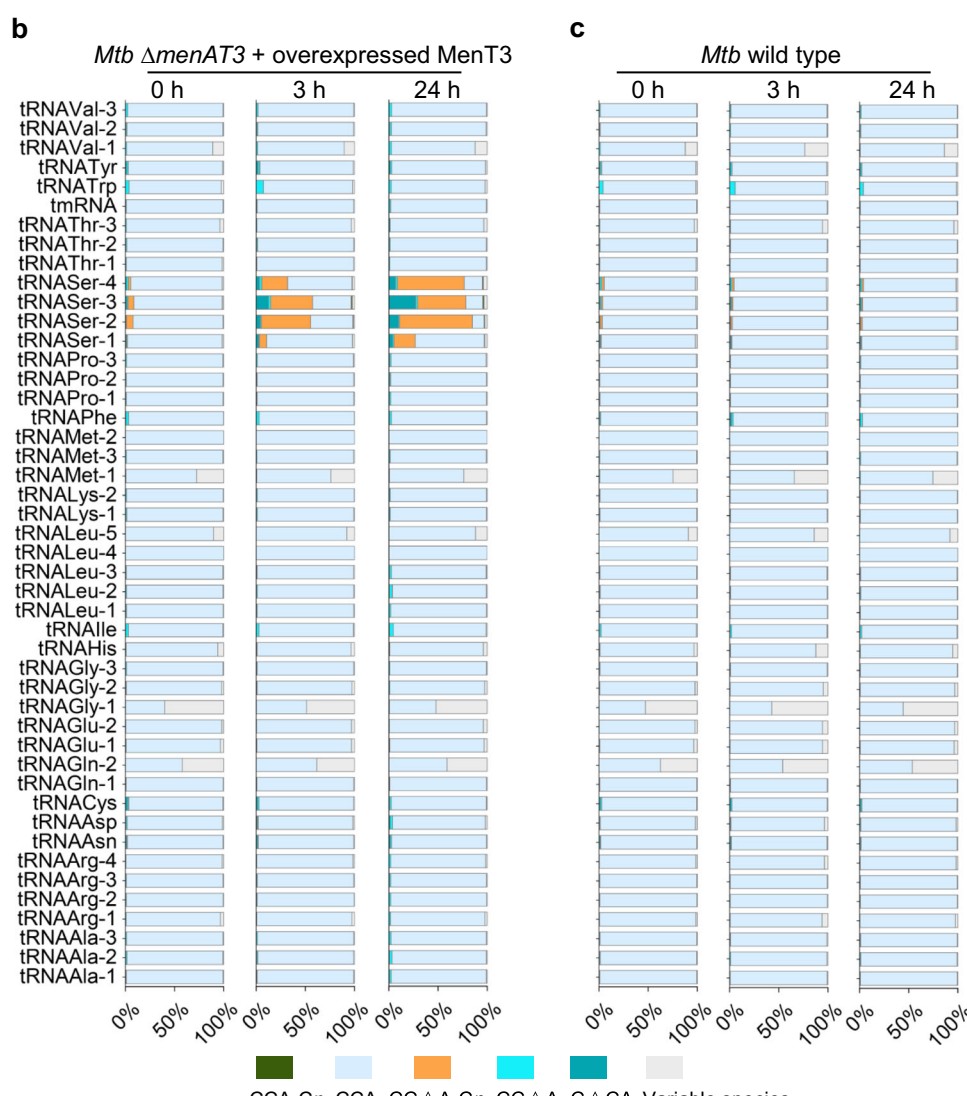

**Fig. 4 | MenT3 specifically targets tRNA^Ser in *M. tuberculosis*. a** Experimental conditions for tRNA-seq in *M. tuberculosis*. *M. tuberculosis* (*Mtb*) wild-type H37Rv strain and its isogenic mutant Δ*menAT3* expressing MenT3 from the integrative pGMC vector were individually grown at 37 °C in 7H9 medium supplemented with 10% albumin-dextrose-catalase (ADC, Difco) and 0.05% Tween 80. When the OD₆₀₀ reached to about 0.5, the anhydrotetracycline inducer (Atc, 200 ng ml⁻¹) was added, and cells were collected after 0, 3, or 24 h incubation at 37 °C. Total RNA was extracted, and tRNA-seq was performed. **b** Percentage of modified tRNA per tRNA species identified for the mutant overexpressing MenT3 and panel **c** for the wild type strain. The names of the identified tRNA for both strains are shown on the left of (**b**). The data were presented as the mean value obtained from three independent experiments. A detailed view of the different modifications obtained for tRNA^Ser is shown in Supplementary Fig. S3. Source data are provided as a Source Data file.

(Fig. 5b). Remarkably, co-overexpression of PcnA also led to a significant increase in the truncated forms of tRNA^Ser-3 (Fig. 5b). Accordingly, tRNA-seq analysis further confirmed that overexpression of PcnA (but not RpH) in the presence of MenT3 leads to the reappearance of CCA ends in all four tRNA^Ser (Supplementary Fig. 4a), which is in agreement with the growth rescue by PcnA (Fig. 5a). In addition, tRNA-seq also confirmed the increase accumulation of tRNA^Ser with ΔA and ΔCA 3′-ends, as observed in *M. tuberculosis* (Supplementary Fig. 4a). Although not present in *M. tuberculosis*, we observed that tRNA^Sec (targeted by MenT3 in *M. smegmatis*) was not affected by PcnA

overexpression. To further study the impact of CCA-adding enzyme on MenT3 in vitro, we purified PcnA of *M. tuberculosis* and first showed that it could efficiently perform tRNA^Ser-4 ΔCCA maturation in vitro in the presence of NTPs, as expected from previous work (Fig. 5c)[27]. We next asked whether the appearance of the truncated forms of tRNA^Ser accumulating in the presence of MenT3 could be generated by PcnA pyrophosphorolysis in vitro. To test this, we purified MenT3-modified tRNA^Ser-4 as performed with RpH (Supplementary Fig. S5) and incubated it with PcnA in the presence of PPi, as described in refs. 32,33. The results from Fig. 5c show that PcnA is indeed capable of generating

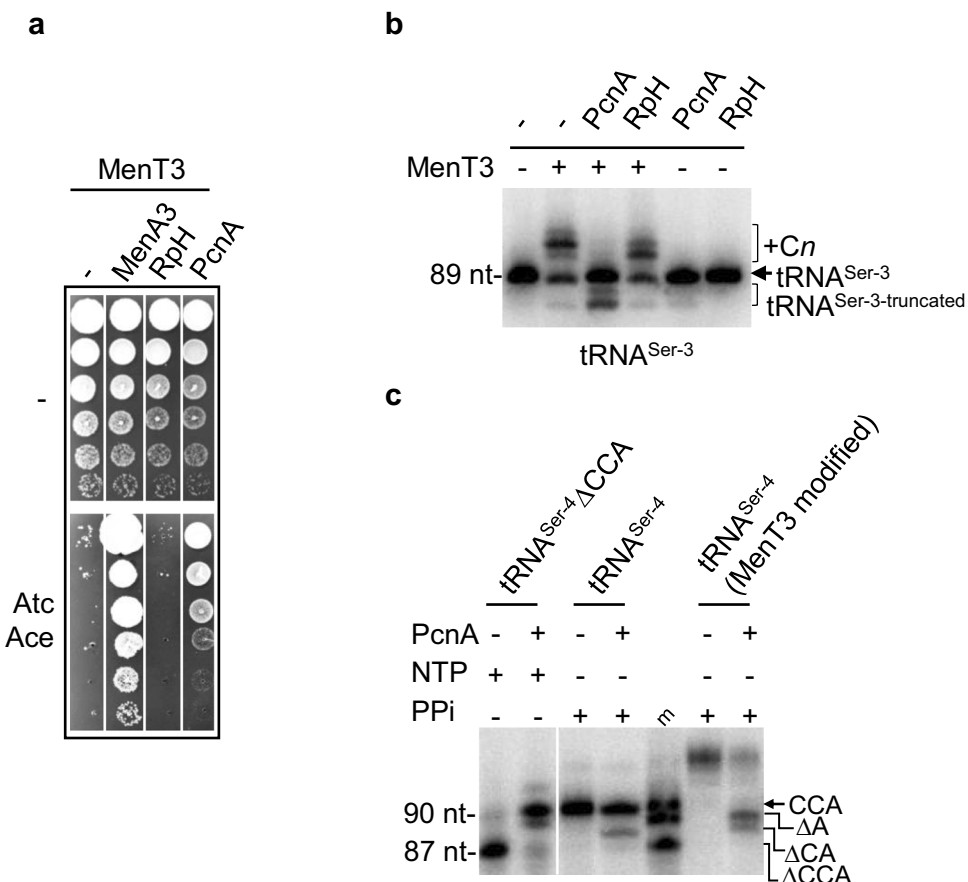

**Fig. 5 | CCA-adding enzyme PcnA counteracts MenT3. a** Suppression of MenT3 toxicity in *M. smegmatis* by the antitoxin MenA3 (control) and PcnA. *M. smegmatis* strain containing the plasmid pGMC-MenT3 was transformed with pLAM12-based plasmids harboring RpH or PcnA of *M. smegmatis*. Cultures were serially diluted and spotted on LA plates with or without inducers (Atc, 100 ng ml⁻¹; Ace, 0.01%). Plates were incubated for 3 days at 37 °C. Shown are representative results from triplicate experiments. **b** Northern blot analysis of RNA extracts from *M. smegmatis* co-transformed with various plasmid combinations: pGMC/pLAM, pGMC-MenT3/pLAM, pGMC-MenT3/pLAM-PcnA, pGMC-MenT3/pLAM-RpH, pGMC/pLAM-RpH, and pGMC/pLAM-PcnA. Cultures of transformants were grown until an OD600 of 0.1 in fresh LB medium and induced with 100 ng ml⁻¹Atc and 0.2% Ace for 3 h at 37 °C. RNA was extracted and analyzed by Northern blot to detect the presence of specific tRNA^Ser species. Shown are representative results from triplicate experiments. **c** In vitro activity of the *M. tuberculosis* PcnA enzyme. tRNA^Ser-4ΔCCA was incubated with 0.5 μM PcnA in the presence of 1 mM NTP at 37 °C for 10 min to test the CCA-addition nucleotidyltransferase (NTase) activity. To examine whether PcnA can counteract MenT3 by its pyrophosphorolysis activity, tRNA^Ser-4 and MenT3-modified tRNA^Ser-4 were incubated with 0.5 μM PcnA in the presence of 1 mM inorganic pyrophosphate (PPi) at 37 °C for 10 min. Shown are representative results from triplicate experiments. Source data are provided as a Source Data file.

truncated tRNA^Ser-4ΔA and ΔCA from MenT3-modified tRNA. Together with the resistance to MenT3 conferred by PcnA, these data suggest that PcnA is likely responding to the toxin by trimming the primary MenT3-modified 3′CCA+C*n* ends of tRNA^Ser in order to regenerate either 3′CCA or compatible ends for repair.

## Steady-state tRNA^Ser modification in *M. tuberculosis*
The 3′-OH tRNA-seq analysis of the *M. tuberculosis* H37Rv wild-type control under normal growth conditions (Supplementary Fig. S3) revealed a significant steady-state level of tRNA^Ser 3′-ends with MenT3-like additions (about 8% of the total tRNA^Ser). In this case, the only possible source of MenT3 would come from the native chromosomal *menAT3* operon (Supplementary Fig. S3). In order to investigate whether endogenous *menAT3* from wild-type *M. tuberculosis* could indeed control the active pool of tRNA^Ser, we performed similar 3′-OH tRNA-seq experiments using the *M. tuberculosis* H37Rv wild type, its isogenic Δ*menAT3* mutant and the Δ*menAT3* mutant carrying MenT3 on the pGMC integrative plasmid in the absence of inducer, in order to obtain low-level leaky expression of MenT3 and avoid toxicity. The data presented in Fig. 6 and Supplementary Fig. S6 show that indeed, the MenT3-like modifications of tRNA^Ser that are present in *M. tuberculosis*

wild type decrease significantly in the Δ*menAT3* mutant. Note that MenT3 expression without inducer was sufficient to partly restore tRNA^Ser alteration in a Δ*menAT3* mutant background, yet less efficiently than in the presence of inducer (Fig. 6 compared to Fig. 4b, c). When comparing each of the tRNA^Ser isoacceptors (Supplementary Fig. S6), the MenT3-like alterations in the Δ*menAT3* mutant dropped from 9.5 to 1.8% for tRNA^Ser-2, from 10.5 to 3% for tRNA^Ser-3, from 9.2 to 4.1% for tRNA^Ser-4 and from 6.7 to 3.8% for tRNA^Ser-1, which was also less efficiently modified following MenT3 overexpression (see Fig. 4c). Among these, the 3′CCΔA+C*n* ends of tRNA^Ser were barely detected in the Δ*menAT3* mutant (Supplementary Fig. S6). Together these data indicate that there is a fraction of MenT3 toxin that is active under standard laboratory growth conditions that can modulate the pool of mature tRNA^Ser available for translation in wild-type *M. tuberculosis*.

## Discussion
This work demonstrates that the MenT3 toxin has a robust NTase activity in vitro and identifies its native tRNA targets in *M. tuberculosis*. It shows that MenT3 can modify all the tRNA tested in vitro by specifically adding cytidines to their 3′-end, although with a strong preference for tRNA^Ser, to which significantly longer stretches of cytidines

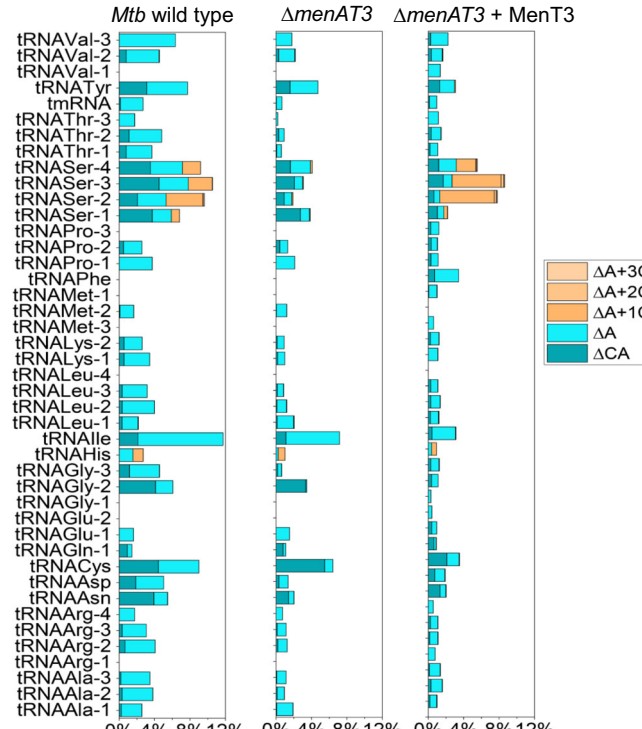

**Fig. 6 | Steady-state tRNA^ser modification in *M. tuberculosis* by endogenous MenT3.** *M. tuberculosis* wild-type H37Rv, the isogenic *ΔmenAT3* mutant and *ΔmenAT3*-menT3 complementation strain were individually grown at 37 °C in 7H9 medium supplemented with 10% albumin-dextrose-catalase (ADC, Difco) and 0.05% Tween 80. When the $OD_{600}$ reached to 0.5, the cells were collected. Total RNA was extracted to perform tRNA-seq. The percentage of modified tRNA per tRNA species identified is shown. The data were presented as the mean value obtained from three independent experiments. A detailed view of the different species obtained for tRNA^Ser is shown in Supplementary Fig. S6. Source data are provided as a Source Data file.

observations indicate that the selective targeting of a single tRNA or a small subset of tRNAs may be a hallmark of successful toxins of TA systems.

This work also shows the importance of the length of the long variable loop characteristic of tRNA^Ser for MenT3 activity[44], which is in agreement with recent work on MenT3[25]. Noticeably, this region was shown to be important for binding to the Seryl-tRNA synthetase (SerRS) both in *E. coli* and in humans[44,45], suggesting that it could indeed contribute to the specific binding of MenT3. Importantly, our work also reveals that the orientation of the variable loop plays a critical role in discriminating tRNA^Ser from other class II tRNA (e.g., tRNA^Leu), which possess similar long variable loop[25].

We observed striking differences in the nature of the tRNA^Ser 3′-end species that accumulated in *M. tuberculosis* when compared to in vitro, i.e., mainly CCΔA + C*n* and CΔCA (with a small fraction of CCΔA, ΔCCA, and CCA + C*n*), and exclusively CCA + C*n* in vitro. This strongly suggests that cytidine elongations are recognized and trimmed by *M. tuberculosis* tRNA maturation/repair enzymes, although not enough to restore a sufficient pool of mature tRNA. Accordingly, we have identified the essential class II CCA-adding enzyme PcnA[27] as the main protein responsible for such tRNA species accumulation. Indeed, PcnA was capable of generating CCΔA and CΔCA ends from MenT3-modified tRNA^Ser in vitro and to efficiently counteract MenT3 toxicity when overexpressed, mainly producing wild-type CCA ends. Yet, both CCΔA and CΔCA end species were also detected under such conditions, which is reminiscent of the 3′-end species identified in the presence of overexpressed MenT3 in the wild-type strain.

These data suggest a model in which MenT3 would first attack uncharged tRNA^Ser 3′-end, adding cytidine extensions (CCA+C*n*) to block aminoacylation. Such MenT3-modified tRNAs would represent bona fide clients for PcnA and perhaps other maturation/repair enzymes, i.e., RpH[46], which can trim 3′-ends to regenerate CCA, but also CCΔA and CΔCA in vitro. Since CΔCA is not a substrate of MenT3, such tRNA species may thus increasingly accumulate as a product of PcnA attempts to restore tRNA 3′-ends. In contrast, both CCA and CCΔA 3′-ends will be modified by MenT3 to generate CCA +C*n* and CCΔA+C*n*, thus progressively leading to an accumulation of CCΔA + C*n* and CΔCA species. Therefore, under the constant pressure of MenT3 activity, the endogenous PcnA might be overwhelmed by the new pool of accumulating CCΔA + C*n* and CΔCA tRNA^Ser species, and thus unable to efficiently regenerate mature CCA ends compatible for further aminoacylation. Whether both PcnA and MenT3 simultaneously or sequentially bind to tRNA^ser remains to be determined.

Our data highlight the existence of a fraction of tRNA^Ser that is modified by MenT3 in *M. tuberculosis* wild type under standard growth conditions. This unexpected discovery suggests the existence of a population of endogenous MenT3 that is active and that could modulate translation, as part of a significant contribution to the control of pathogen growth during infection. In agreement with this hypothesis, a screen for transposon mutants that failed to grow in murine macrophages identified *menT3*, and not *menA3*, as one of the 126 genes important for *M. tuberculosis* intracellular survival under such conditions[47]. More recently, *menT3* was also identified as a persistent gene in *M. tuberculosis*, which contributes to the long-term survival in mouse lungs[48], and a double *menT3/T4* deletion strain was shown to be attenuated and to confer protection in mice and guinea pigs models[49], thus further supporting the requirement of endogenous MenT3 activity under relevant physiological conditions for this pathogen.

The fact that MenT3 can be active and target tRNA^Ser despite the presence of the MenA3 antitoxin is intriguing and might be related to its peculiar mode of inhibition. Indeed, previous work showed that MenA3 acts as a type VII antitoxin, which likely inhibits MenT3 by stimulating its auto-phosphorylation at the S78 catalytic site

were added. Transcriptomic identification of MenT3-tRNA targets in *M. tuberculosis* revealed that tRNA^Ser isoacceptors are the sole targets of the toxin and that significantly different tRNA 3′-ends accumulate in vitro, most likely due to detoxification attempts by the essential CCA-adding enzyme PcnA. Finally, this work identifies a basal level of MenT3-dependent tRNA^Ser 3′-end alterations in *M. tuberculosis* under standard growth conditions, when MenT3 levels are physiological and produced from the native chromosomal *menAT3* operon.

The observation that overexpressed MenT3 only targets tRNA^Ser isoacceptors and that cytidine elongations are significantly shorter (1 to 5) than in vitro (up to 17) suggests more stringent conditions in *M. tuberculosis* or/and that other factors would contribute to such a strong preference. Remarkably, tRNA^Ser were recently shown to be significantly less aminoacylated than other tRNAs in *M. tuberculosis*, with less than 20% of tRNA^Ser being charged when at steady state[34]. Such a low charging of tRNA^Ser was proposed to be due to a competition between the aminoacylation of tRNA^Ser for translation and the need for serine amino acids for pyruvate and acetate production and glycine synthesis in *E. coli*[35]. Therefore, the low availability of endogenous charged tRNA^Ser in *M. tuberculosis* would further exacerbate the deleterious effect of MenT3. The targeting of specific tRNAs is a characteristic of many toxin families[34,36–43]. In *M. tuberculosis*, the acetyltransferase toxin TacT (Rv0919) was shown to specifically acetylate the primary amine group of charged tRNA^Gly glycyl-tRNAs[34], and the PIN domain RNase toxin VapC-mt4 was shown to cleave a single site within the anticodon sequence of tRNA^Cys [15]. Collectively, these

residue[21,23,50]. Accordingly, this inhibitory mechanism was recently supported by phospho-proteome analysis of *M. tuberculosis*, which indeed identified the presence of phosphorylated MenT3 at S78[51]. Therefore, one attractive possibility is that a fraction of inactive phosphorylated MenT3 could be dephosphorylated in response to certain growth conditions or aggression by the host immune system. In support of this, the housekeeping phosphatase PstP of *M. tuberculosis* could dephosphorylate MenT3 in vitro[23]. Reversibly, a fraction of free active MenT3 could also be affected by endogenous Ser/Thr protein kinases (STPKs) like PknD and PknF[51], suggesting that the level of active endogenous MenT3 toxin might not be solely be controlled by the anti-toxin, but also by responsive endogenous kinase/phosphatase networks to ensure translational control is attuned to physiological need during growth and infection.

## Methods

### Bacterial strains
*E. coli* strains DH5α (Invitrogen) and BL21 (λDE3) AI (Novagen), *M. tuberculosis* H37Rv (WT; ATCC 27294) and its mutant derivative H37Rv Δ*menAT3::dif6/pGMCZ* and *M. smegmatis* mc²155 (ATCC 700084) have been described[21]. *E. coli* were grown at 37 °C in LB, when necessary, with kanamycin (Km, 50 µg ml⁻¹), ampicillin (Ap, 50 µg ml⁻¹), ʟ-arabinose (ʟ-ara, 0.1% w/v) or ᴅ-glucose (glu, 0.2% w/v). *M. smegmatis* mc²155 was grown at 37 °C in LB, when necessary, with Km (10 µg ml⁻¹) or streptomycin (Sm, 25 µg ml⁻¹). *M. tuberculosis* strains were grown at 37 °C in 7H9 medium (Middlebrook 7H9 medium, Difco) supplemented with 10% albumin-dextrose-catalase (ADC, Difco) and 0.05% Tween 80 (Sigma-Aldrich), or on complete 7H11 solid medium (Middlebrook 7H11 agar, Difco) supplemented with 10% oleic acid-albumin-dextrose-catalase (OADC, Difco). When necessary, media were supplemented with, hygromycin (Hy, 50 µg m⁻¹), Sm (25 µg ml⁻¹), zeocin (Zeo, 25 µg ml⁻¹), acetamide (Ace, 0.01%), or anhydrotetracycline (Atc, 100 or 200 ng ml⁻¹)[20].

### Bacterial growth assay
Toxicity assay in *M. smegmatis* was performed as described[20,52]. Briefly, cultures of mc²155 strain grown in LB medium at 37 °C were co-transformed with the integrative pGMC-MenT3 and with pLAM12-vector, pLAM-MenA3, pLAM-RpH, pLAM-PcnA, or pLAM-PcnA DLD (59-61)-AAA, and selected on LB Km Sm agar plates for 3 days at 37 °C in the presence or absence of Atc (100 ng ml⁻¹) and Ace (0.01%) to induce expression of the toxin and the suppressor candidates, respectively.

### Plasmid constructs
Plasmids pET15b (Novagen), pET20b, pGMCS, pLAM12, and pETDuet-1 have been described. All the primers used to construct the plasmids are described in Supplementary Data 1.

To construct pET15b-MenT3, *rv1045* was PCR amplified from the *M. tuberculosis* H37Rv genome using primers Rv1045 NdeI-For and Rv1045 BamHI-Rev and cloned into pET15b after digestion with NdeI and BamHI enzymes. For pET20b-mtbRpH, *rv1340* was PCR amplified from the *M. tuberculosis* H37Rv genome using primers Rv1340 NdeI-For and Rv1340 XhoI-Rev. Amplified fragments were then inserted into pET20b following digestion with NdeI and XhoI enzymes. To construct pETDuet-1-mtbPcnA, *rv3907c* was PCR amplified from the *M. tuberculosis* H37Rv genome using primers Rv3907c In-Fusion-For and Rv3907c In-Fusion-Rev, and cloned by homologous recombination in linearized pETDuet-1 plasmid using In-Fusion HD Cloning Kits (Takara Bio). Plasmids pGMC-MenT3 and pLAM-MenA3 have been described[21]. Plasmids pLAM-RpH and pLAM-PcnA were obtained following PCR amplification of MSMEG_4901 with primers MSMEG_4901 NdeI-For and MSMEG_4901 EcoRI-Rev, MSMEG_6926 with primers MSMEG_6926 NdeI-For and MSMEG_6926 EcoRI-Rev, cloned as NdeI/EcoRI fragments into NdeI/EcoRI digested pLAM12, respectively. pLAM-PcnA

DLD (59-61)-AAA was constructed by Quik-Change site-directed mutagenesis using appropriate primers and pLAM-PcnA as template. The pUC57-T7-Met-2-HDV, pUC57-T7-His-HDV, pUC57-T7-Leu-3-HDV, and pUC57-T7-Ser-4-HDV plasmids containing tRNA-HDV fusion under the control of a T7 promoter were synthesized by Genewiz (Azenta Life Sciences). The pUC57-T7-Met-2+variable loop of Ser-4-HDV, pUC57-T7-Ser-4ΔA-HDV, pUC57-T7-Ser-4ΔCA-HDV, pUC57-T7-Ser-4ΔCCA-HDV, and pUC57-T7-Ser-4Δvariable loop-HDV, pUC57-T7-Ser-4ΔCA in VL-HDV, pUC57-T7-Ser-4ΔCA and G-C pair in VL-HDV, pUC57-T7-Ser-4ΔCA, G-C, and G-C pair in VL-HDV, pUC57-T7-Ser-4ΔCA C-G, G-C and G-C pair in VL-HDV, and pUC57-T7-Ser-4 + GU-HDV, and pUC57-T7-Leu-3ΔGU-HDV plasmids were constructed by PCR using the primers listed in Supplementary Data 1. The sequences of the T7-His-HDV, T7-Leu-3-HDV, T7-Met-2-HDV, T7-Met-2 + variable loop of Ser-4-HDV, T7-Ser-4-HDV, and T7-Ser-4Δvariable loop-HDV fragments are given in Supplementary Data 1. To prepare the DNA template for transcription with T7 polymerase, T7-tRNA-HDV fragments were PCR amplified using primers T7-For and HDV-Rev for T7-Met-2-HDV and T7-His-HDV, or T7-For and HDV short-Rev for T7-Met-2+variable loop of Ser-4-HDV, T7-Ser-4-HDV, T7-Leu-3-HDV, T7-Ser-4Δvariable loop-HDV, T7-Ser-4ΔA-HDV, T7-Ser-4ΔCA-HDV, T7-Ser-4ΔCCA-HDV, T7-Ser-4ΔCA in VL-HDV, T7-Ser-4ΔCA, and G-C pair in VL-HDV, T7-Ser-4ΔCA, G-C and G-C pair in VL-HDV, T7-Ser-4ΔCA C-G, G-C, and G-C pair in VL-HDV, T7-Ser-4 + GU-HDV, T7-Leu-3ΔGU-HDV was constructed by In-fusion PCR and the primers were listed in Supplementary Data 1.

### Protein expression and purification
Purified MenT3 was produced as described previously[21]. Purified proteins were also obtained as follows: to purify MenT3, PcnA, RpH of *M. tuberculosis*, strain BL21(λDE3) AI transformed with pET15b-MenT3, pETDuet-1-mtbPcnA, or pET20b-mtbRpH was grown to an OD₆₀₀ of approximately 0.4 at 37 °C, 0.2% ʟ-ara was added and the culture immediately incubated overnight at 22 °C, respectively. Cultures were centrifuged at 5000×*g* for 10 min at 4 °C, pellets were resuspended in lysis buffer (50 mM Tris-HCl, pH 8.0, 200 mM NaCl, 10 mM MgCl₂, 20 mM imidazole; 20 ml for 1 liter of cell culture) and incubated for 30 min on ice. Lysis was performed using the One-shot cell disrupter at 1.5 Kbar (One-shot model, Constant Systems Ltd). Lysates were centrifuged for 30 min at 30,000×*g* at 4 °C, and the resulting supernatants were gently mixed with Ni-NTA Agarose beads (Qiagen) pre-equilibrated with lysis buffer, at 4 °C for 30 min in a 10 ml poly-prep column (Bio-Rad). The column was then stabilized for 10 min at 4 °C, washed five times with 10 ml of lysis buffer, and proteins were eluted with elution buffer (50 mM Tris-HCl, pH 8.0, 200 mM NaCl, 10 mM MgCl₂, 250 mM imidazole). About 500 µl elution were collected and PD MiniTrap G-25 columns (GE Healthcare) were used to exchange buffer (50 mM Tris-HCl, pH 8.0, 200 mM NaCl, 10 mM MgCl₂, 10% glycerol) and proteins were concentrated using vivaspin 6 columns with a 5000 Da cut off (Sartorius). To remove the His-tag, thrombin was incubated with the protein at 4 °C overnight. Following NTA and streptavidin addition, the cleaved His-tag and thrombin were washed out. Proteins were stored at −80 °C until further use.

### In vitro transcription of tRNAs
tRNAs were synthesized via in vitro transcription using PCR templates that incorporated an integrated T7 RNA polymerase promoter sequence. Primers for *M. tuberculosis* tRNAs are given in Supplementary Data 1. The T7 RNA polymerase in vitro transcription reactions were carried out in a total volume of 25 µl, which included a 5 µl nucleotide mix containing 2.5 mM NTPs (Promega). For each reaction, 50 ng to 100 ng of template DNA were used, along with 1.5 µl of rRNasin (40 U.ml⁻¹, Promega), 5 µl of 5x optimized transcription buffer (Promega), 2 µl of T7 RNA polymerase (20 U ml⁻¹), and 2.5 µl of 100 mM DTT. The reactions were incubated at 37 °C for 2 h[21]. The resulting

tRNA products were extracted using Trizol reagent[53] and stored at a final concentration of 100 to 200 ng µl⁻¹, as determined by NanoDrop analysis. To obtain high-concentration tRNA for EMSA, the TranscriptAid T7 High Yield Transcription Kit from Thermo Fisher was employed. In this protocol, 1 µg of tRNA DNA template was included in the transcription assay and incubated at 37 °C for 4 h. Subsequently, DNase I treatment was performed, followed by RNA isolation using Trizol.

### In vitro transcription of tRNAs with homogeneous 3'-ends

An optimized version of the hepatitis delta virus (HDV) ribozyme was used to generate homogeneous tRNA 3'-ends as described[20,54]. Briefly, the DNA template T7-tRNA-HDV was amplified from plasmid pUC-57Kan-T7-tRNA-HDV (Supplementary Data 1). Labeled or unlabeled tRNAs were prepared by in vitro transcription of PCR templates using T7 RNA polymerase. The T7 RNA polymerase in vitro transcription reactions were performed in 25 µl total volume, with a 5 µl nucleotide mix of 2.5 mM ATP, 2.5 mM UTP, 2.5 mM GTP, 60 µM CTP (Promega, 10 mM stock) and 2–4 µl 10 µCi µl⁻¹ of radiolabelled CTP [α-32P], or with 5 µl nucleotide mix of 2.5 mM ATP, 2.5 mM UTP, 2.5 mM GTP, 2.5 mM CTP for unlabeled tRNA transcription. 50 to 100 ng of template were used per reaction with 1.5 µl rRNasin 40 U ml⁻¹ (Promega), 5 µl 5x optimized transcription buffer (Promega), 2 µl T7 RNA polymerase (20 U ml⁻¹) and 2.5 µl 100 mM DTT. Unincorporated nucleotides were removed by Micro Bio-Spin 6 columns (Bio-Rad) according to the manufacturer's instructions. The transcripts were gel-purified on a denaturing 6% acrylamide gel and eluted in 0.3 M sodium acetate overnight at 20 °C. The supernatant was removed, ethanol precipitated, and resuspended in 14 µl nuclease-free water. Radioactively labeled tRNAs carrying a 2',3' cyclic phosphate at the 3'-end was dephosphorylated using T4 polynucleotide kinase (NEB) in 100 mM Tris-HCl pH 6.5, 100 mM magnesium acetate and 5 mM β-ME in a final volume of 20 µl for 6 h at 37 °C. All assays were desalted by Micro Bio-Spin 6 columns (Bio-Rad).

### Nucleotide transfer assay

MenT3 NTase activity was assayed in 10 µl reaction volumes containing 20 mM Tris-HCl pH 8.0, 10 mM NaCl, 10 mM MgCl₂, and 1 µCi µl⁻¹ of radiolabelled rNTPs [α-32P] (Hartmann Analytic) and incubated for 5 min at 37 °C. About 100 ng in vitro transcribed tRNA product, 1 µg total RNA, or 100 ng of E. coli or M. smegmatis tRNA was used per assay with 0.2 µM of protein. The 10 µl reactions were purified with Bio-Spin® 6 Columns (Bio-Rad), and mixed with 10 µl of RNA loading dye (95% formamide, 1 mM EDTA, 0.025% SDS, xylene cyanol and bromophenol blue), denatured at 90 °C and separated on 6% polyacrylamide-urea gels. The gel was vacuum dried at 80 °C, exposed to a phosphorimager screen, and revealed by autoradiography using a Typhoon phosphorimager (GE Healthcare).

For the single tRNA 3' elongation by MenT3 in vitro, the radiolabeled tRNA was incubated with 0.02 or 0.2 µM MenT3 in the presence of 1 mM CTP for 2 or 5 min at 37 °C, as indicated. The reaction was then halted by adding 10 µl of RNA loading dye, denatured at 90 °C, and RNA samples were separated on a 10% polyacrylamide-urea gel.

### tRNA repair in vitro

The tRNA-modified product from MenT3 in vitro reactions underwent the following steps: it was initially purified using Bio-Spin® 6 Columns (Bio-Rad), followed by incubation with RpH (10 µM) in the reaction buffer containing 50 mM Tris-HCl, pH 8.0, 10 mM NaCl, 2.5 mM MgCl₂, 10 mM K₂HPO₄, 1 mM DTT at 37 °C for an indicative time. Subsequently, it was mixed with 10 µl of RNA loading dye, subjected to denaturation at 90 °C, and then separated using 10% polyacrylamide-urea gels. Alternatively, the RNA samples were analyzed through RNA-seq.

### Activity assay for CCA-adding enzyme PcnA

PcnA NTase activity was assayed in 10 µl reaction volumes containing 20 mM Tris-HCl pH 8.0, 10 mM NaCl, 10 mM MgCl₂, 1 mM NTP, radiolabeled tRNA^Ser-4^ΔCCA tRNA, and 0.5 µM protein, incubated for 10 min at 37 °C. Pyrophosphorolysis was conducted at 37 °C in the same buffer, except NTP was replaced by 1 mM PPi. Radiolabeled tRNA^Ser-4^ or MenT3-modified tRNA^Ser-4^ was tested in the assay. Subsequently, reaction mixtures were combined with 10 µl of RNA loading dye, denatured at 90 °C, and resolved on 10% polyacrylamide-urea gels.

### Northern blot

Total RNA was isolated from M. smegmatis transformed with pGMC-MenT3 or plasmids as indicated in figure legends. Subsequently, 5 µg of RNA was separated using 10% polyacrylamide-urea gels. Following gel electrophoresis, the RNA was transferred onto nylon membranes (Amersham Hybond N+, GE Healthcare) and subjected to hybridization as previously described[55]. The sequences of the oligonucleotides used are described in Supplementary Data 1.

### Alphafold3 model building and refinement

To study the interaction between MenT3 and tRNA, the sequences of MenT3 and tRNA^Ser-4^, tRNA^Leu-3^, or tRNA^Leu-3^ΔGU were submitted to the AlphaFold3 server[56], incorporating Mg²⁺ ions. The pTM and ipTM scores were used to assess the quality of complex predictions as described by Alphafold3. The output structural models were then analyzed using PREDICTOMES to evaluate interaction models between MenT3 and the tRNA variable loop. Confidence in the models was assessed using predicted local-distance difference test (pLDDT) scores, with higher scores indicating more reliable predictions engaging in higher confidence interactions. Visualization of the interaction models was performed using ChimeraX[57], where potential hydrogen bonds were predicted with the 'hbonds' command set to a distance tolerance of 1 Å and an angle tolerance of 20°, and represented as black dashed lines in the figures.

### tRNA libraries and sequencing

Primers used for the construction of tRNA libraries are described in Supplementary Data 1. To obtain the MenT3 library from in vitro reactions, 5 µg of M. smegmatis total RNA supplemented with 1 mM CTP was incubated with water and 0.8 µg MenT3, then incubated for 20 min at 37 °C. For the in vitro transcribed tRNA-seq, 20 ng µL⁻¹ of specific tRNAs were incubated with 1 µM MenT3 at 37 °C for 10 min. Total RNA samples and single tRNA samples were isolated using trizol and ethanol precipitation, respectively. Construction of tRNA-seq libraries from M. smegmatis and M. tuberculosis were performed as follows: M. smegmatis transformed with pGMC or pGMC-menT3 weak RBS were grown at 37 °C for 3 days, then the culture was transferred to fresh LB, until OD₆₀₀ reached 0.1, Atc (100 ng ml⁻¹) was added, and the cells were collected after 3 h at 37 °C for preparation of total RNA, as previously described[20]. M. tuberculosis wild-type H37Rv, H37Rv ΔmenAT3 mutant, or the same strains transformed with pGMC-menT3 weak RBS were grown at 37 °C until OD₆₀₀ reached 0.5, Atc (200 ng ml⁻¹) was added, and the cells were collected at 0, 3, or 24 h. Cell pellets were resuspended in 1 ml of Trizol, and cells were disrupted in a bead-beater disrupter, after the addition of glass beads. Samples were centrifuged for 2 min at 20,000×g, and the trizol extract was collected and conserved for at least 48 h at −80 °C before transferring out of the BSL3 laboratory for total RNA isolation. All of the RNA was dissolved in DEPC-H₂O (pH 7.0) by heat at 65 °C for 10 min, which conditions known to be able to discharge the tRNA in vitro (Walker & Fredrick, 2008). To remove m1A, m3C, and m1G modifications in the tRNAs, the total RNA samples were pretreated using the demethylase kit (Arraystar, cat#: AS-FS-004, Rockville, MD, USA), followed by trizol isolation. Notably, during the demethylation reaction (pH 7.5–8)[58,59],

tRNA was deacylated sufficiently by this near-neutral pH. The 3p-v4 oligo was 5′ adenylated using a 5′ DNA Adenylation Kit (E2610S, NEB) according to the manufacturer's protocol. To construct the library, RNA samples were ligated to the adenylated 3p-v4 adapters, and reverse transcription was performed with ProtoScript II RT enzyme (NEB) using barcode primers (Supplementary Data 1). Finally, PCR amplification was performed with tRNAs oligoFor mix and A-PE-PCR10 (after 5 cycles, the program was paused, and B_i7RPI1_CGTGAT or i7RPI7_GATCTG was added) using Q5 Polymerase Hot-Start (NEB). The library was sequenced by DNBSEQ -G400RS High-throughput Sequencing Set (PE150) in BGI Genomics (Hong Kong).

### RNA-Seq data processing

After a quality check, reads were demultiplexed to obtain a fastq file per experimental condition. For each experimental condition, the same procedure was applied: (i) mapping to a reference, (ii) PCR duplicate removal, and (iii) quantification of read counts. Raw reads quality was checked with FastQC (http://www.bioinformatics.babraham.ac.uk/projects/fastqc). Further read processing was performed with R software version 4.3.1 and BioConductor libraries for processing sequencing data obtained. R library ShortRead 1.58.0[60] was used to process fastq files. Rsubread 2.14.2[61] was used to map reads on the reference genome. Further filtering was performed to ensure reads contained the structure resulting from library preparation (Supplementary Data 1 for RNA-Seq read structure): reads start with a random nucleotide at position 1, a valid barcode at position 2–5, a recognition sequence resulting from ligation at position 6–24 with one mismatch allowed, random UMI sequence at position 25–39, agacat control sequence at position 40–45, and a random nucleotides sequence at position 46–50, which corresponds to the reverse complement of the ligated 3′-end of tRNAs in the experiment. Reads corresponding to the different experimental conditions identified by the barcode were demultiplexed into fastq files for independent further analyses (mapping and quantification).

All the *M. smegmatis* mc²155 (NC_008596.1) and *M. tuberculosis* H37Rv (NC_000962.3) chromosomal tRNA gene sequences were extracted in a multifasta file. The nucleotides CCA were concatenated at the end of each sequence. Mapping on reference sequences was performed by using the library Rsubread 2.14.2 with parameters ensuring that the read maps without any indel to a unique region: unique=TRUE, Type='dna', maxMismatches=2, indels=0, ouput_format = "BAM".

For quantification, SAM files were processed with Rsamtools 2.16.0 (https://bioconductor.org/packages/Rsamtools) to count the number of reads mapping to the same region in reference sequences with the same 3′ unmapped sequences due to RNA 3′-end alterations. Reads were kept if the aligned region nearest the 3′ end of the reference sequence was at least 20 nucleotides long and allowed 3′ end soft clipping (unaligned region after the minimum 20 nucleotides aligned nearest the 3′ end of the reference sequence), which corresponds to nucleotides added by the toxin. In the alignments in the SAM file, this translates to a CIGAR code ending with only 'M' characters (indicating a match), possibly followed by "S" characters (indicating no alignment to the reference sequence). PCR duplicates were removed using the UMI introduced in the library preparation for reads mapping exactly the same region (same beginning and end of the alignment) and having exactly the same unmapped 3′-end sequence. After these preprocessing steps, reads were regrouped by their 3′-end mapping position and unmapped 3′-end region sequence to quantify their abundance in terms of the same tRNA 3′-end followed by the same nucleotides i.e., same post-transcriptional modifications. To remove noise, a tRNA with its post-transcriptional modifications was kept only if it was found at least ten times in at least one of the experimental conditions.

Primers and specific sequences used in this work are shown in Supplementary Data 1.

### Reporting summary

Further information on research design is available in the Nature Portfolio Reporting Summary linked to this article.

## Data availability

The RNA-seq data generated in this study have been deposited in the NCBI database under accession code PRJNA1171215. All the data generated in this study are provided in the Supplementary Information/Source Data file. Source data are provided with this paper.

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

## Acknowledgements

We thank Ciarán Condon, Simon Lebaron, and Leonora Poljak for the useful discussion. This work was supported by the Centre National de la Recherche Scientifique, Université Paul Sabatier, the Program d'Investissements d'Avenir (ANR-20-PAMR-0005) to O.N. and P.G.; the Swiss

National Science Foundation (CRSII3_160703) to P.G.; the National Natural Science Foundation of China, (32000021) to X.X.; a scholarship from the China Scholarship Council (CSC) as part of a joint international PhD program with -Toulouse University Paul Sabatier; a Fondation pour la Recherche Médicale (FDT202304016729) to X.H. and (EQU202403018015) to P.G.; a Springboard Award (SBF002\1104) from the Academy of Medical Sciences to B.U. and T.R.B.; an Engineering and Physical Sciences Research Council Molecular Sciences for Medicine Centre for Doctoral Training studentship (EP/S022791/1) to T.J.A. For the purpose of open access, the author has applied a Creative Commons Attribution (CC BY) license to any Author Accepted Manuscript version arising from this submission.

## Author contributions

Analyzed data: X.X., R.B., B.V., T.J.A., B.U., C.G., X.H., P.R., T.R.B., O.N., and P.G. Designed research: X.X., R.B., B.V., T.J.A., B.U., C.G., X.H., C.P., P.R., T.R.B., O.N., and P.G. Performed research: X.X., R.B., B.V., T.J.A., B.U., C.G., X.H., and C.P. Wrote the paper: X.X. and P.G. with contributions from all the authors. Funding acquisition: X.X., O.N., T.R.B., and P.G. Supervised the study: P.G.

## Competing interests

The authors declare no competing interests.
