## [Peer Review File · Nature Communications]

REVIEWER COMMENTS

Reviewer #1 (Remarks to the Author):

Xu et al. described the substrate specificity of Mtb toxin MenT3 in both in vitro and in vivo. They discovered that MenT3 adds cytidine to most tRNA species in the in vitro reaction system. In particular, multiple tRNA-Ser species showed extended addition of cytidine compared to other tRNA species. A long variable loop of tRNA-Ser is necessary and sufficient for the extension of long cytidine at the 3' end of tRNAs. In contrast, tRNA-seq of cellular tRNA profile showed that MenT3 overexpression exclusively adds cytidine to tRNA-Ser species, and the length of C stretch is shorter than the C stretch added by recombinant MenT3 protein in vitro. Authors showed that RNase PH can trim the C stretch at the 3' end of tRNA-Ser in the in vitro condition. Based on this observation, the authors claim that RNase PH counteracts MenT3 in the cell. Individual experiments are well-designed and performed. However, this reviewer is not fully convinced of the authors' conclusions due to the discrepancy between in vitro and in vivo results regarding substrate specificity of MenT3 and a lack of evidence that RNase PH and/or other factors counteract MenT3 in the cell.

Major points

1. As mentioned above, MenT3 has a broad substrate specificity and ability to add long stretch to tRNA-Ser in vitro, while the cellular tRNA profile showed that strict substrate specificity to tRNA-Ser and the length of C stretch is shorter. As the authors admit, these conflicts are likely due to the presence of additional cellular factors that modulate MenT3 activity. The authors claim that the presence of RNase PH in the cell is the cause of shorter C stretch. However, there is no evidence that shows RNase PH counteracts MenT3 in the cell. Since in vitro condition does not fully recapture the in vivo condition, in vivo evidence is necessary to support their conclusion. To test this hypothesis, authors should
 - a. Test whether deletion or overexpression of RNase PH facilitates or rescues the growth arrest caused by MenT3 overexpression.
 - b. If #a is true, test whether deletion or overexpression of RNase PH affects the profile of 3' cytidine extension in the cell.
2. As the authors discussed aminoacylation can affect MenT3 activity. The variable loop of tRNA-Ser is recognized by SerRS, so it is possible that MenT3 competes against SerRS. Since aminoacylated tRNA cannot be the substrate for MenT3, it is necessary to address the possibility that aminoacylation protects tRNA from MenT3. Similar to RNasePH, the authors should test whether overexpression of SerRS affects MenT toxicity and profile of cytidine addition in the cell.
3. Some other tRNAs have long variable loops, including tRNA-Leu and tRNA-Tyr. Experiments addressing how MenT3 distinguishes long variable loops of tRNA-Ser from other tRNAs species is critical for the authors claim that MenT3 recognizes long variable loop in tRNA-Ser.

Minor points

1. The use of tRNA modification as cytidine addition is confusing because tRNA modification also means chemical modification of nucleosides.
2. L230 should be Fig 4

Reviewer #2 (Remarks to the Author):

The manuscript by Xu and colleagues describes the activity of the Mycobacterium tuberculosis MenT3 toxin. Manuscript contributes the precise identification of the targets of MenT3 toxin in vivo and most importantly explains the mechanism of the intoxication. The intoxication results not only from the MenT3 toxin, but the final products are also a result of the cellular activity of ribonuclease PH. Together the two enzymes generate a spectrum of dead-end products, unsuitable for tRNA aminoacylation. Authors also detect a background activity of the MenT3 toxin in vivo without the overexpression. This is an interesting observation suggesting that some of the toxin could be constantly released from the toxin-antitoxin complex. Overall, the manuscript provides original observations that are important in the field of toxins as well as they could be eventually relevant for studies of M. tuberculosis physiology or infection. Manuscript is very well written and easy to follow, the provided research is very systematic and exhaustive and casts no doubts. I only have a few minor comments.

General questions:

I am missing some more information on the ribonuclease pH. What is the natural role of this enzyme? It seems that the role of RpH would be to restore the tRNA acceptor stems, but it often goes too far and chews up the CCA end. Has this activity been previously reported in the literature? Or is it MenT3-specific?

While MenT3 was found to highly specifically modify the tRNASer upon overexpression in vivo (Figure 3b), the deletion of menAT3 seems to result in a slight decrease of modification of most of the tRNA, not only the tRNASer (Figure 5a). Is there any possible explanation for this effect?

Minor observations for the text:

Line 22 – I think we lack sufficient proof to conclude that T-A systems “control” bacterial growth. Despite a few examples, that are debated, I think this generalization is an overstatement. Also, since they are not the core genes, and it is not clear whether their activation or inactivation should be more detrimental, I doubt that they are attractive therapeutic targets. Nevertheless, I think that the research on T-As is interesting, and I would suggest choosing more neutral introduction.

Line 33-34 – I am not sure that the background activity of the MenT3 necessarily implicates its importance in the infection – is there any data showing that delta MenT3 compromised for the infection?

Line 46, It would be more precise to say “under steady state conditions” instead of “in the absence of stress” because in the loss of plasmid, which is one of the examples mentioned in the following sentence regarding the activation of T-As get activated, is not really a stress condition per se.

All of the sources cited at line 65 come from the same laboratory, with multiple expressions of concern. Among the three citations here, Agarwal et al., 2018 article has two corrigenda published followed by the editorial expression of concern. Tiwari et al., 2015 also has a corrigendum. I leave it up to authors to decide whether they want to refer to these works.

Line 130-2 – according to labeling of tRNAs used throughout the text, it should be Ser-2 and Ser-3.

Line 176 – should be “was replaced”

Line 182 – I am wondering if the word “activity” is correct in this case. My interpretation is that variable loop of the target Serine tRNAs is rather responsible for the processivity of the MenT3 enzyme, not the activity per se.

Reviewer #3 (Remarks to the Author):

The article titled “Nucleotidyltransferase toxin MenT targets and extends the aminoacyl acceptor ends of serine tRNAs in vivo to control Mycobacterium tuberculosis growth” elucidates the in vivo targets of the MenT3 toxin, a well-established nucleotidyltransferase by the same research group. By employing 3'-OH tRNA sequencing, the authors validate tRNA^{Ser} as the exclusive substrate of MenT3 in vivo. Additionally, the study demonstrates that Mtb RNase PH effectively mitigates the toxicity resulting from MenT3 overexpression in Mycobacterium tuberculosis. Overall, the article confirms the in vivo targets of MenT3 and corroborates earlier in vitro findings by the same group. A recent publication by Liu et al. in NAR, 2024, also offers insights into the functional and structural characteristics of MenT3. It will be important for the authors to juxtapose their findings with these latest discoveries to highlight similarities and differences.

In a previous article by the same research group (Cai et al., Science Advances, 2020), tRNA^{Ser} was clearly identified as the preferred substrate of MenT3 using in vitro transcribed tRNAs from Mycobacterium tuberculosis as well as E. coli. The earlier study also highlights E. coli RNase PH's ability to counteract the toxicity induced by MenT3, emphasizing pronounced toxicity by MenT3 and rescue by Mtb RNase PH in Mtb cells. However, the current article extends these in vitro observations to an in vivo context and falls short of exploring the physiological significance of these modifications.

A major problem that I had with the manuscript is a clear summary of prior art and what this manuscript adds to that existing knowledge. In my opinion, the article does not add significant new insights to be considered for publication in a top journal like Nature communications

Major points:

- tRNA^{Ser} Aminoacylation Levels:

- o The authors mention that tRNA^{Ser} aminoacylations are very low (approximately 20%) and refer to an earlier paper. However, the cited paper relies on an old article that does not explain the observed low levels of aminoacylation. It's important to note that tRNA^{Ser} typically exhibits low aminoacylation levels (approximately 60-80%) in bacteria, but not as low as the ~20% observed in Mtb. To clarify this discrepancy, the authors should provide steady-state levels of tRNA^{Ser} in vivo via northern blot analysis.

- Importance of Variable Loop in tRNA^{Ser}:

- o Upon examining Figure 2E, the claim made in lines 171-173 that “the deletion of the variable loop of tRNA^{Ser}-4 eliminates the formation of long poly-C extensions” appears to be unsubstantiated. To validate this assertion, a detailed quantification of the activity difference is required.

- o Considering the quantification data mentioned above, it is essential for the authors to explain whether the significance of the variable loop is solely attributed to a gain of function, rather than a loss of function. This distinction is crucial for understanding the role of the variable loop.

o If the variable loop is indeed important, as stated by the authors, then why do different tRNASer molecules exhibit significant differences in their modification status? Further clarification is needed to address this discrepancy.

o To explore substrate specificity further, the authors need to analyze the unique nucleotide bases of tRNASer in comparison with tRNA^{Leu} and tRNA^{Tyr}, both of them also possess long variable arms.

- Role of RpH in Recycling Modified tRNAs:

o Line 215 highlights the role of RpH in recycling the MenT3-modified tRNAs. However, does overexpression of RpH effectively rescue Mtb toxicity?

o In lines 217-218, the authors embark on studying three candidate RNases. Instead, they could have employed a homology-based search using *E. coli* RNase PH as a query. It would be valuable to understand the homology of these three RNases with respect to *E. coli* RNase PH.

- Steady state tRNASer modification in *M. tuberculosis*:

o Author state in line 257-259 that “there is a fraction of MenT3 toxin that is active under standard laboratory growth conditions that can modulate the pool of mature tRNASer available for translation in wild type *M. tuberculosis*”. Does this level change with multiple stages of Mtb life cycle? The physiological significance of this needs to be explained.

Minor points:

- In line 169: mentioned reference does not show the structural differences in tRNAs

- In abstract, line 31-32 should be modified as “only in the presence of the native menAT3 operon.

- In supplementary S1: Source of tRNAs is missing

- In supplementary S1: The figure creates confusion as the ‘+’ and ‘-’ symbol denoting the presence and absence of MenT3 in the biochemical reaction is haphazard.

- In method section: method details are missing for removing CTP from reaction. CTP removal is mentioned in line 220 of result section.

- In method section: throughout methods units like ng.ml⁻¹ should be ng ml⁻¹. Space should be used instead of dot.

- In line 176: replace should be replaced

- In line 232: comma is missing after thus

- In line 307: amino acylation should be aminoacylation

- In method section for EMSA: Percentage of acrylamide used is missing.

- In line 493: transfer should be transferring

- In line 542: i.e. should be i.e.,

- References are not uniform with missing volume/issue/page numbers at multiple places and organism names are not italicised. In line 686, what star indicates?

Ref: NCOMMS-24-07692

Title: Nucleotidyltransferase toxin MenT targets and extends the aminoacyl acceptor ends of serine tRNAs in vivo to control *Mycobacterium tuberculosis* growth

Response to Reviewers

We would like to thank the Reviewers for their positive comments and constructive criticism. Point-by-point responses to Reviewers' comments are listed below in blue, and where necessary, changes have been made to the original manuscript and are shown in yellow highlight. In some instances, changes have been made to figures but these are not altered in color, in order to maintain consistency within the figure.

REVIEWER COMMENTS

Reviewer #1 (Remarks to the Author):

Xu et al. described the substrate specificity of Mtb toxin MenT3 in both in vitro and in vivo. They discovered that MenT3 adds cytidine to most tRNA species in the in vitro reaction system. In particular, multiple tRNA-Ser species showed extended addition of cytidine compared to other tRNA species. A long variable loop of tRNA-Ser is necessary and sufficient for the extension of long cytidine at the 3' end of tRNAs. In contrast, tRNA-seq of cellular tRNA profile showed that MenT3 overexpression exclusively adds cytidine to tRNA-Ser species, and the length of C stretch is shorter than the C stretch added by recombinant MenT3 protein in vitro. Authors showed that RNase PH can trim the C stretch at the 3' end of tRNA-Ser in the in vitro condition. Based on this observation, the authors claim that RNase PH counteracts MenT3 in the cell. Individual experiments are well-designed and performed. However, this reviewer is not fully convinced of the authors' conclusions due to the discrepancy between in vitro and in vivo results regarding substrate specificity of MenT3 and a lack of evidence that RNase PH and/or other factors counteract MenT3 in the cell.

We thank the reviewer for the positive evaluation of our work.

Major points

1. As mentioned above, MenT3 has a broad substrate specificity and ability to add long stretch to tRNA-Ser in vitro, while the cellular tRNA profile showed that strict substrate specificity to tRNA-Ser and the length of C stretch is shorter. As the authors admit, these conflicts are likely due to the presence of additional cellular factors that modulate MenT3 activity. The authors claim that the presence of RNase PH in the cell is the cause of shorter C stretch. However, there is no evidence that shows RNase PH counteracts MenT3 in the cell. Since in vitro condition does not fully recapture the in vivo condition, in vivo evidence is necessary to support their conclusion. To test this hypothesis, authors should
 - a. Test whether deletion or overexpression of RNase PH facilitates or rescues the growth arrest caused by MenT3 overexpression.
 - b. If #a is true, test whether deletion or overexpression of RNase PH affects the profile of 3' cytidine extension in the cell.

A combined response to comments 1a and 1b (both concerning RpH) is provided below:

As proposed, we have tested experimentally whether RpH overexpression could rescue growth in response to MenT3 in mycobacteria, as we previously observed in *E. coli* (Cai et al., 2020). *M. smegmatis* RpH was cloned on a multicopy plasmid and expressed under the control of a strong acetamide inducible promoter. In sharp contrast with *E. coli*, RpH had no detectable inhibitory effect on MenT3 toxicity in *M. smegmatis* (new Fig. 5A). In addition, northern blot analysis of tRNA^{Ser} performed *in vivo* following RpH co-overexpression revealed that only a small fraction of CMP extensions was shortened (new Fig. 5B), but not restored, indicating that RpH activity is not the main response to MenT3 modification *in vivo* in mycobacteria. These new data, together with the previous observation that MenT3 is much more toxic in *M. smegmatis* and *M. tuberculosis* than in *E. coli* stirred us to search for other mycobacterial factors involved in responding to MenT3 activity.

Noticeably, only few tRNA genes of *M. smegmatis* and *M. tuberculosis* encode a CCA triplet at their 3' termini, which is in sharp contrast with *E. coli*, in which all the tRNA genes possess a CCA triplet. Accordingly, tRNA maturation in *M. tuberculosis* requires the essential CCA-adding enzyme PcnA, which is not the case in *E. coli*. Strikingly, we found that overexpression of PcnA confers resistance to MenT3 in mycobacteria (new Fig. 5A; new Supplementary Figure S4C). In addition, we show by northern blot and tRNA-seq that the suppressive effect was associated with the appearance of predominant tRNA^{Ser} wild-type forms and the disappearance of the CMP elongations induced by MenT3 (new Fig. 5B, new Supplementary Figure S4A). Remarkably, co-overexpression of PcnA also led to a significant increase in the truncated forms of tRNA^{Ser}. Such effect was recapitulated *in vitro* with purified PcnA (new Fig. 5C).

These new data suggest that PcnA can efficiently counteract MenT3 activity both *in vivo* and *in vitro*.

To accompany the new figures, the following text has been added on pages 13 and 14 (line 265 to 303):

“Yet, in sharp contrast with *E. coli*, overexpression of RpH from *M. smegmatis* failed to suppress MenT3 toxicity in *M. smegmatis* (Fig. 5A). In addition, although RpH overexpression in the presence of MenT3 could generate shorter tRNA^{Ser} (CCΔA+C_n) ends, it did not lead to a detectable accumulation of 3' CCA, CCΔA or CΔCA species *in vivo* (Fig. 5B). This suggests that although RpH is capable of targeting MenT3 modified tRNA^{Ser}, it is not the main factor responsible for the accumulation of the truncated tRNA species identified *in vivo* (Fig. 4).

Only few tRNA genes of *M. smegmatis* and *M. tuberculosis* encode a CCA triplet at their 3' termini, *i.e.*, 14 out of 46 and 13 out of 45, respectively (Błaszczuk *et al.*, 2023). This is in sharp contrast with *E. coli*, in which all the tRNA genes possess such triplet (Zhu & Deutscher, 1987). Accordingly, tRNAs maturation in *M. tuberculosis* required the essential CCA-adding enzyme PcnA (DeJesus *et al.*, 2017; Błaszczuk *et al.*, 2023), while it is not the case in *E. coli* (Deutscher *et al.*, 1977). In addition to tRNA maturation, CCA-adding enzymes also contribute to tRNA quality control by targeting tRNA for degradation (Wilusz *et al.*, 2011), or during CCA synthesis, by excision of the 3' terminal nucleotide in the presence of inorganic pyrophosphate (PPi) *in vitro* (Igarashi *et al.*, 2011; Hager *et al.*, 2022). Strikingly, we found that overexpression of PcnA in *M.*

smegmatis confers resistance to MenT3 (Fig. 5A). As a control, mutation in the NTase activity center of PcnA abolished such suppressive effect, indicating that active PcnA enzyme is required for growth rescue (Supplementary Figure S4C). Northern blot analysis of tRNA^{Ser-3} following co-expression of PcnA and MenT3 confirmed such effect, as judged by the appearance of predominant tRNA^{Ser-3} wild-type forms and the disappearance of the Cn elongations induced by MenT3 (Fig. 5B). Remarkably, co-overexpression of PcnA also led to a significant increase in the truncated forms of tRNA^{Ser-3} (Fig. 5B). Accordingly, tRNA-seq analysis further confirmed that overexpression of PcnA (but not RpH) in the presence of MenT3 leads to the reappearance of CCA ends in all four tRNA^{Ser} (Supplementary Figure 4A), which is in agreement with the growth rescue by PcnA (Fig. 5A). In addition, tRNA-seq also confirmed the increase accumulation of tRNA^{Ser} with ΔA and ΔCA 3'-ends, as observed *in vivo* in *M. tuberculosis* (Supplementary Figure 4A). Although not present in *M. tuberculosis*, we observed that tRNA^{Sec} (targeted by MenT3 in *M. smegmatis*) was not affected by PcnA overexpression. To further study the impact of CCA-adding enzyme on MenT3 *in vitro*, we purified PcnA of *M. tuberculosis* and first showed that it could efficiently perform tRNA^{Ser-4} ΔCCA maturation *in vitro* in the presence of NTPs, as expected from previous work (Fig. 5C) (Błaszczuk *et al*, 2023). We next asked whether the appearance of the truncated forms of tRNA^{Ser} accumulating in the presence of MenT3 *in vivo* could be generated by PcnA pyrophosphorolysis *in vitro*. To test this, we purified MenT3-modified tRNA^{Ser-4} as performed with RpH (Supplementary Figure S5) and incubated it with PcnA in the presence of PPi, as described (Igarashi *et al*, 2011; Hager *et al*, 2022). The results from Fig. 5C show that PcnA is indeed capable of generating truncated tRNA^{Ser-4} ΔA and ΔCA from MenT3 modified tRNA. Together with the *in vivo* resistance to MenT3 conferred by PcnA, these data suggest that PcnA is likely responding to the toxin by trimming the primary MenT3-modified 3'CCA+Cn ends of tRNA^{Ser} in order to regenerate either 3'CCA or compatible ends for repair.”.

2. As the authors discussed aminoacylation can affect MenT3 activity. The variable loop of tRNA-Ser is recognized by SerRS, so it is possible that MenT3 competes against SerRS. Since aminoacylated tRNA cannot be the substrate for MenT3, it is necessary to address the possibility that aminoacylation protects tRNA from MenT3. Similar to RNasePH, the authors should test whether overexpression of SerRS affects MenT toxicity and profile of cytidine addition in the cell.

As requested, in parallel with RpH and PcnA, we have tested co-overexpression of *M. smegmatis* SerRS with MenT3, but we could not detect any suppressive effect (see Figure #R1 not included in the manuscript). Although we cannot exclude that SerRS could somehow compete with MenT3, it does not appear to be a major contributor *in vivo*.

Figure #R1: *M. smegmatis* co-expressing MenT3 from a pGMC anhydrotetracycline-inducible vector and SerRS (or the antitoxin MenA3 control) from a pLAM acetamide-inducible vector were serially diluted and spotted on LB agar plates with or without inducers. Plates were incubated for 3 days at 37°C.

3. Some other tRNAs have long variable loops, including tRNA-Leu and tRNA-Tyr. Experiments addressing how MenT3 distinguishes long variable loops of tRNA-Ser from other tRNA species is critical for the authors' claim that MenT3 recognizes long variable loop in tRNA-Ser.

We have extensively addressed this specific issue in our response to Reviewer 3 comment #2

Minor points

1. The use of tRNA modification as cytidine addition is confusing because tRNA modification also means chemical modification of nucleosides.

When possible, the term tRNA modification has been changed, although we had to keep it at some places when a mix of C addition and Δ CA or Δ A are considered.

2. L230 should be Fig 4 This has been modified.

Reviewer #2 (Remarks to the Author):

The manuscript by Xu and colleagues describes the activity of the *Mycobacterium tuberculosis* MenT3 toxin. Manuscript contributes the precise identification of the targets of MenT3 toxin *in vivo* and most importantly explains the mechanism of the intoxication. The intoxication results not only from the MenT3 toxin, but the final products are also a result of the cellular activity of ribonuclease PH. Together the two enzymes generate a spectrum of dead-end products, unsuitable for tRNA aminoacylation. Authors also detect a background activity of the MenT3 toxin *in vivo* without the overexpression. This is an interesting observation suggesting that some of the toxin could be constantly released from the toxin-antitoxin complex. Overall, the manuscript provides original observations that are important in the field of toxins as well as they could be eventually relevant for studies of *M. tuberculosis* physiology or infection. Manuscript is very well written and easy to follow, the provided research is very systematic and exhaustive and casts no doubts. I only have a few minor comments.

We thank the reviewer for the positive evaluation of our work.

General questions:

I am missing some more information on the ribonuclease pH. What is the natural role of this enzyme? It seems that the role of RpH would be to restore the tRNA acceptor stems, but it often goes too far and chews up the CCA end. Has this activity been previously reported in the literature? Or is it MenT3-specific?

Previous work has shown that RpH could indeed restore tRNA acceptor stems and also trim further the CCA triplet *in vitro* (Wen et al 2005), which is now cited line 379, page 19.

While MenT3 was found to highly specifically modify the tRNA^{Ser} upon overexpression *in vivo* (Figure 3b), the deletion of *menAT3* seems to result in a slight decrease of modification of most of the tRNA, not only the tRNA^{Ser} (Figure 5a). Is there any possible explanation for this effect?

We agree with the reviewer that the slight decrease in \$\Delta A\$ and \$\Delta CA\$ detected in the *menAT3* mutant is intriguing. Although very weak, the absence of the MenA3 antitoxin (a transcriptional regulator as well as antitoxin against MenT3) could well be indirectly responsible for such an effect. In support of such hypothesis, it was recently shown that a *menT3/T4* double mutation affects a significant number of transcripts in *M. tuberculosis*, including other toxin genes (Gosain, T.P., Chugh, S., Rizvi, Z.A. et al. *Mycobacterium tuberculosis* strain with deletions in *menT3* and *menT4* is attenuated and confers protection in mice and guinea pigs. Nat Commun 15, 5467 (2024). [https://doi-org.insb.bib.cnrs.fr/10.1038/s41467-024-49246-5](https://doi.org/insb.bib.cnrs.fr/10.1038/s41467-024-49246-5)). The activity of MenA antitoxins in the absence of their cognate toxin genes is beyond scope of the current study though could be pursued in future.

Minor observations for the text:

Line 22 – I think we lack sufficient proof to conclude that T-A systems “control” bacterial growth. Despite

a few examples, that are debated, I think this generalization is an overstatement. Also, since they are not the core genes, and it is not clear whether their activation or inactivation should be more detrimental, I doubt that they are attractive therapeutic targets. Nevertheless, I think that the research on T-As is interesting, and I would suggest choosing more neutral introduction.

Accordingly, we have now stated that toxins of TA systems can inhibit and not control growth and deemphasized the attractive therapeutic targets from the abstract.

Line 33-34 – I am not sure that the background activity of the MenT3 necessarily implicates its importance in the infection – is there any data showing that delta MenT3 compromised for the infection?

Yes, there are currently several interesting papers from three independent groups that point towards a role for *menT3* deletion in mice, macrophages and guinea pigs, which is not common for toxin mutants. This is why we strongly believe that the unexpected background activity in lab conditions could become highly relevant in the infection context.

See adapted discussion on page 19 line 392-399.

“In agreement with this hypothesis, a screen for transposon mutants that failed to grow in murine macrophages identified *menT3*, and not *menA3*, as one of the 126 genes important for *M. tuberculosis* intracellular survival under such conditions (Rengarajan *et al*, 2005). More recently, *menT3* was also identified as a persistence gene in *M. tuberculosis*, which contributes to the long term survival in mouse lungs (Dutta *et al*, 2014) and a double *menT3/T4* deletion strain was shown to be attenuated and to confer protection in mice and guinea pigs models (Gosain *et al*, 2024), thus further supporting the requirement of endogenous MenT3 activity under relevant physiological conditions for this pathogen”.

Line 46, It would be more precise to say “under steady state conditions” instead of “in the absence of stress” because in the loss of plasmid, which is one of the examples mentioned in the following sentence regarding the activation of T-As get activated, is not really a stress condition per se.

This has been changed accordingly.

All of the sources cited at line 65 come from the same laboratory, with multiple expressions of concern. Among the three citations here, Agarwal *et al*, 2018 article has two corrigenda published followed by the editorial expression of concern. Tiwari *et al*, 2015 also has a corrigendum. I leave it up to authors to decide whether they want to refer to these works.

We agree with the reviewer and we have now removed these references from the manuscript.

Line 130-2 – according to labeling of tRNAs used throughout the text, it should be Ser-2 and Ser-3.

This has been changed accordingly.

Line 176 – should be “was replaced”

This has been changed accordingly.

Line 182 – I am wondering if the word “activity” is correct in this case. My interpretation is that variable loop of the target Serine tRNAs is rather responsible for the processivity of the MenT3 enzyme, not the activity per se.

We agree with the reviewer and this has been replaced by “modification by MenT3”.

Reviewer #3 (Remarks to the Author):

The article titled “Nucleotidyltransferase toxin MenT targets and extends the aminoacyl acceptor ends of serine tRNAs in vivo to control Mycobacterium tuberculosis growth” elucidates the in vivo targets of the MenT3 toxin, a well-established nucleotidyltransferase by the same research group. By employing 3'-OH tRNA sequencing, the authors validate tRNA^{Ser} as the exclusive substrate of MenT3 in vivo. Additionally, the study demonstrates that Mtb RNase PH effectively mitigates the toxicity resulting from MenT3 overexpression in Mycobacterium tuberculosis. Overall, the article confirms the in vivo targets of MenT3 and corroborates earlier in vitro findings by the same group. A recent publication by Liu et al. in NAR, 2024, also offers insights into the functional and structural characteristics of MenT3. It will be important for the authors to juxtapose their findings with these latest discoveries to highlight similarities and differences.

In a previous article by the same research group (Cai et al., Science Advances, 2020), tRNA^{Ser} was clearly identified as the preferred substrate of MenT3 using in vitro transcribed tRNAs from Mycobacterium tuberculosis as well as E. coli. The earlier study also highlights E. coli RNase PH's ability to counteract the toxicity induced by MenT3, emphasizing pronounced toxicity by MenT3 and rescue by Mtb RNase PH in Mtb cells. However, the current article extends these in vitro observations to an in vivo context and falls short of exploring the physiological significance of these modifications.

A major problem that I had with the manuscript is a clear summary of prior art and what this manuscript adds to that existing knowledge. In my opinion, the article does not add significant new insights to be considered for publication in a top journal like Nature communications.

We thank the reviewer for his/her appreciation of our work.

Yet, with all our respect we do not agree about the reduced novelty of this work. Instead, we believe that this is the first work that provides an exhaustive analysis of the tRNA that are targeted by MenT NTase toxins *in vivo* in its native bacteria, namely *M. tuberculosis*. Note that there is no MenT toxin identified in *E. coli* and that MenT are much less toxic in *E. coli* than in mycobacteria (Cai et al., 2020), indicating that performing such studies in mycobacteria is important. In addition, this work also shows for the first time that in contrast with what was previously published (Cai et al., 2020; Liu et al., 2024), the 3' ends accumulating *in vivo* following MenT3 expression in *M. tuberculosis* are very different than

the one produced *in vitro*. Importantly, this *in vivo* work also reveals for the first time the existence of a background level of toxin activity under normal growth conditions, without overexpression of the toxin. This suggests that MenT3 toxin could be constantly liberated from its antagonistic antitoxin *in vivo*. We strongly believe that such observations are important in the field of toxins and highly relevant for studies of *M. tuberculosis* physiology and infection.

We have now made significant efforts to experimentally address the reviewer's comments below.

Major points:

- tRNA^{Ser} Aminoacylation Levels:

- o The authors mention that tRNA^{Ser} aminoacylations are very low (approximately 20%) and refer to an earlier paper. However, the cited paper relies on an old article that does not explain the observed low levels of aminoacylation. It's important to note that tRNA^{Ser} typically exhibits low aminoacylation levels (approximately 60-80%) in bacteria, but not as low as the ~20% observed in Mtb. To clarify this discrepancy, the authors should provide steady-state levels of tRNA^{Ser} *in vivo* via northern blot analysis.

Such a hypothesis, which is only stated in our discussion, points towards the likely possibility that the low tRNA^{Ser} charging previously observed in *M. tuberculosis* could contribute to the fact that tRNA^{Ser} is a good target for active MenT3 *in vivo* in *M. tuberculosis*. Such hypothesis is based on very recent work by Rubin's lab (mBio 2023) who show that less than 20% of tRNA^{Ser} are charged at steady state in *M. tuberculosis*. Therefore, we believe that it is reasonable to refer to this specific work as a hypothesis, not having to repeat similar experiments, also in *M. tuberculosis*, which we believe would be out of the scope of this study.

The following text have been modified in line 341:

“Remarkably, tRNA^{Ser} were recently shown to be significantly less aminoacylated than other tRNAs in *M. tuberculosis*, with less than 20% of tRNA^{Ser} being charged when at steady state (Tomasi *et al*, 2023)”.

- Importance of Variable Loop in tRNA^{Ser}:

- o Upon examining Figure 2E, the claim made in lines 171-173 that “the deletion of the variable loop of tRNA^{Ser}-4 eliminates the formation of long poly-C extensions” appears to be unsubstantiated. To validate this assertion, a detailed quantification of the activity difference is required.

Our data on the impact of deletion of the long variable loop (VL) of tRNA^{Ser} is in line with the data recently published by Liu *et al*. in NAR 2024, which provides kinetics of CMP incorporation of wild-type and mutant tRNA^{Ser} showing that the VL deletion dramatically decreased its CMP incorporation by MenT3. This work has now been discussed at several places in our revised version (see below). To further validate such activity difference, we have now tested different MenT3 concentrations and provide a quantification of the percentage of modified tRNA^{Ser} wild-type and Δ VL, showing a dramatic decrease in modification when the VL was absent.

The following text and figures (**new supplementary figure S2A**) have been added on page 9 line 174:

“Note that in contrast with tRNA^{Ser-4}, the Δ VL tRNA^{Ser-4} variant deleted of the variable loop was not detectably modified by MenT3 when lower MenT3 concentrations were used (**Supplementary Figure S2A**)”.

Considering the quantification data mentioned above, it is essential for the authors to explain whether the significance of the variable loop is solely attributed to a gain of function, rather than a loss of function. This distinction is crucial for understanding the role of the variable loop.

A more detailed analysis of the role of the variable loop is provided in our response to the next comments. In addition, the recent biochemical and structural work Liu *et al.*, in NAR 2024 strongly suggests that by interacting with the N-terminal region of MenT3, the long variable loop of tRNA^{Ser} could serve as an anchor that would allow the proper locating the CCA 3' end into to the active site of MenT3, thus facilitating efficient CMP incorporation.

The following sentence has been added line 184, page 10:

“Together these data suggest that the variable loop is critical for tRNA^{Ser} 3'-end CMP addition and elongation by MenT3. Accordingly, a recent study showed the variable loop of tRNA^{Ser} might indeed act as an anchor that would allow the proper positioning of the CCA 3'-end into to the active site of MenT3, thus facilitating efficient CMP incorporation (Liu *et al.*, 2024)”.

If the variable loop is indeed important, as stated by the authors, then why do different tRNA^{Ser} molecules exhibit significant differences in their modification status? Further clarification is needed to address this discrepancy.

The reviewer refers to the *in vivo* experiments performed in *M. tuberculosis* (Fig. 4), in which we reproducibly found that tRNA^{Ser-1} is proportionally less modified by MenT3 when compared to the other tRNA^{Ser-2, -3, -4} (about 65% of tRNA^{Ser-2, 3, 4} and 20% of tRNA^{Ser-1} as judged from three independent replicates). We currently have no explanation for such a difference, although it is possible that tRNA^{Ser-1} has different modification profile *in vivo* in *M. tuberculosis*, which could affect directly MenT3 binding and modification *in vivo*.

o To explore substrate specificity further, the authors need to analyze the unique nucleotide bases of tRNA^{Ser} in comparison with tRNA^{Leu} and tRNA^{Tyr}, both of them also possess long variable arms.

We agree with the reviewer that this is a very interesting question, especially in the case of tRNA^{Leu-2, 3, 5} of *M. tuberculosis* that have similar variable loop as tRNA^{Ser}.

To answer this, we used tRNA^{Leu-3} as a comparative model because it possesses a long variable loop that is comparable to the one of tRNA^{Ser}.

In order to determine the cause for the observed preference for tRNA^{Ser} over tRNA^{Leu}, we first constructed a tRNA^{Ser-4} chimera containing the loop region of the tRNA^{Leu-3} variable loop (**new Fig. 3B**) and show that the loop region of the variable loop is not responsible for the preference for tRNA^{Ser} over tRNA^{Leu} (**new Fig. 3C**).

We next used this chimera as a template to sequentially delete G/C pairs within the arm of the tRNA^{Ser-4} variable loop (**new Fig. 3B**) and show that the deletion of three G/C pairs within the arm of tRNA^{Ser-4} significantly impaired cytidine addition by MenT3 (**new Fig. 3C; new Supplementary Figure S2C**), indicating that although the length of the arm is important, it is not the only requirement for MenT3's preference for tRNA^{Ser} over tRNA^{Leu}.

Remarkably, tRNA^{Leu-3} contains two extra nucleotides, G60 and U61, between the T-arm and the VL when compared to tRNA^{Ser-4} (**new Fig. 3D**) and the AlphaFold3 model of tRNA^{Leu-3} deleted for nucleotides G60 and U61 shows that the arm of the variable loop is now shifted towards the position of tRNA^{Ser-4} arm (**new Supplementary Figure S2D**). Strikingly, we found that tRNA^{Leu-3} deleted for G60 and U61 is now fully modified by MenT3, in a similar manner as tRNA^{Ser-4} and, conversely, insertion of GU nucleotides between the T-arm and the variable loop of tRNA^{Ser-4} significantly affects modification by MenT3 (**Fig. 3C**). Together these data reveal the importance of the length and the orientation of variable loop of tRNA^{Ser} for MenT3 modification.

The following text has been added, in addition to the **new Fig. 3** and **supplementary Figure S2**, on page 10 line 189:

“Some tRNA in *M. tuberculosis* possess a long variable loop that is comparable to the one of tRNA^{Ser}, especially tRNA^{Leu-2,3 and 5}. Yet, these tRNA are not preferred targets of MenT3 (**Fig. 1E**; (Cai *et al*, 2020). Nevertheless, comparison of the AlphaFold3 models of tRNA^{Leu-3} and tRNA^{Ser-4} in complex with MenT3 shows that the long variable loop region of both tRNA potentially interacts with the N-terminal region of MenT3 (**Fig. 3A**), as recently proposed (Liu *et al*, 2024). However, a significantly more reliable interaction network is predicted in the case of tRNA^{Ser-4} (**Supplementary Figure S2B**). In order to further investigate what determines such a preference for tRNA^{Ser}, we first constructed a tRNA^{Ser-4} chimera containing the loop region of the tRNA^{Leu-3} variable loop (Mut I; **Fig. 3B**) and tested it *in vitro* in the presence of MenT3. Note that low concentrations of the toxin were used in order to better visualize modifications by MenT3 following migration on urea-PAGE gel, as performed in **Supplementary Figure S2A**. The results clearly show that the loop region of the variable loop is not responsible for the preference for tRNA^{Ser} over tRNA^{Leu} (**Fig. 3C**). We next used this chimera as a template to sequentially delete G/C pairs within the arm of the tRNA^{Ser-4} variable loop (Mut II, III, IV; **Fig. 3B**) and tested these constructs *in vitro* in the presence of MenT3. The Mut II construct lacking the first G/C pair, which is equivalent in length to tRNA^{Leu-3}, was still efficiently modified by MenT3, indicating that the difference in length between tRNA^{Ser} and tRNA^{Leu} is also not responsible for MenT3 preference under our experimental conditions. Noticeably, while the deletion of two G/C pairs had no significant effect (Mut III), the deletion of three G/C pairs within the arm of tRNA^{Ser-4} significantly impaired cytidine addition by

Ment3 (Mut IV; **Fig. 3C**; **Supplementary Figure S2C**), indicating that although the length of the arm is important, it is not the only requirement for Ment3 preference. A close-up view of the proposed interaction between the tRNA^{Leu-3} and tRNA^{Ser-4} in complex with Ment3 shows that the arm of the variable loop of tRNA^{Leu-3} is slightly shifted away from the N-terminal region of Ment3 (**Supplementary Figure S2D**), suggesting that the orientation of the variable loop could play a role in Ment3 preference. Remarkably, tRNA^{Leu-3} contains two extra nucleotides G60U61 between the T-arm and the variable loop when compared to tRNA^{Ser-4} (**Fig. 3D**) and the AlphaFold3 model of tRNA^{Leu-3} deleted of nucleotides G60U61 shows that the arm of the variable loop is now shifted towards the position of tRNA^{Ser-4} arm (**Supplementary Figure S2D**). Strikingly, we found that tRNA^{Leu-3} deleted of G60U61 is fully modified by Ment3 in a similar manner as tRNA^{Ser-4} and conversely, insertion of GU nucleotides between the T-arm and the variable loop (at position 62 and 63) of tRNA^{Ser-4} significantly affect modification by Ment3 (**Fig. 3C**). Together these data reveal the importance of the length and the orientation of variable loop of tRNA^{Ser} for Ment3 activity”.

Role of RpH in Recycling Modified tRNAs:

o Line 215 highlights the role of RpH in recycling the Ment3-modified tRNAs. However, does overexpression of RpH effectively rescue Mtb toxicity?

This experiment has been performed: see response to reviewer #1.

o In lines 217-218, the authors embark on studying three candidate RNases. Instead, they could have employed a homology-based search using *E. coli* RNase PH as a query. It would be valuable to understand the homology of these three RNases with respect to *E. coli* RNase PH.

This comment is related to the comments 1 a and b of Reviewer #1.

We now show that in contrast with *E. coli* (Cai *et al.*, 2020), RpH overexpression does not suppress the growth defect induced by Ment3 and does not efficiently restore the tRNA^{Ser} wild type *in vivo*, which suggests that RpH is not responsible for the truncated CCΔA and CΔCA ends accumulating *in vivo* in *M. tuberculosis*. In sharp contrast, we found that the essential CCA-adding enzyme PcnA could efficiently suppress the growth defect induced by Ment3 *in vivo* in mycobacteria. In addition, purified PcnA was capable of trimming CMP stretches from Ment3-modified tRNA^{Ser} *in vitro*, generating CCΔA and CΔCA ends.

These new data are shown in **Fig. 5** and the new text has been added on page 13 and 14 (line 265 to 303). Please see our response to Reviewer #1 comment 1 for a more detailed answer.

• Steady state tRNA^{Ser} modification in *M. tuberculosis*:

o Author state in line 257-259 that “there is a fraction of Ment3 toxin that is active under standard laboratory growth conditions that can modulate the pool of mature tRNA^{Ser} available for translation in

wild type *M. tuberculosis*". Does this level change with multiple stages of Mtb life cycle? The physiological significance of this needs to be explained.

Our finding that a small but detectable pool of modified tRNAs caused by MenT3 activity can be observed *in vivo* under standard laboratory conditions is unprecedented. The result allows us to hypothesise that this pool could increase or decrease in relative size in response to certain stresses, including infection conditions. The fact that mutations in *ment3* are attenuated in infection and that *ment3* was identified as a persistence gene in *M. tuberculosis* is in line with such an hypothesis. An interesting challenge for the future would indeed be to detect conditions in which cytidine modification of tRNA^{Ser} will be associated with specific stress conditions, including the presence of drugs, that are relevant for the tuberculosis pathogen. However, we believe that this should be performed as an independent study dedicated to such questions.

Please see the adapted discussion page 19 line 392-399 (below), discussion on page 20 line 406-414, and our response to comment 4 of Reviewer #2.

Minor points:

- In line 169: mentioned reference does not show the structural differences in tRNAs

M. tuberculosis tRNA sequence alignment pointing specific structural differences are shown in supplementary data Fig. S8 (Cai et al., 2020). This shows that tRNA^{Ser} have a longer VL when compared to other class II tRNA, such as tRNA^{Tyr}, tRNA^{Leu}.

- In abstract, line 31-32 should be modified as "only in the presence of the native menAT3 operon.

This has been changed accordingly.

In supplementary S1: Source of tRNAs is missing

The source of tRNA has now been mentioned in the Figure legend.

- In supplementary S1: The figure creates confusion as the '+' and '-' symbol denoting the presence and absence of MenT3 in the biochemical reaction is haphazard.

This has been changed accordingly.

- In method section: method details are missing for removing CTP from reaction. CTP removal is mentioned in line 220 of result section.

This has now been mentioned it in the method section. The CTP was removed by Bio-Spin® 6 Columns that can be used to clean up RNA samples.

- In method section: throughout methods units like ng.ml-1 should be ng ml-1. Space should be used instead of dot.

This has been changed accordingly throughout the methods section.

- In line 176: replace should be replaced

This has been changed accordingly.

- In line 232: comma is missing after thus

This sentence has been removed from the manuscript in response to Reviewer 2.

- In line 307: amino acylation should be aminoacylation

This has been changed accordingly.

- In method section for EMSA: Percentage of acrylamide used is missing.

Thank you for pointing this mistake. This method was not connected to a current figure and was left in the manuscript by mistake. This has now been deleted.

- In line 493: transfer should be transferring

This has been changed accordingly.

- In line 542: i.e. should be i.e.,

This has been changed accordingly.

- References are not uniform with missing volume/issue/page numbers at multiple places and organism names are not italicised. In line 686, what star indicates?

This has been changed accordingly.

REVIEWERS' COMMENTS

Reviewer #1 (Remarks to the Author):

This reviewer's concern on the original manuscript was a lack of evidence of the author's claim about the counteracting mechanism against MenT3 toxin in which RNase PH regenerate C-added tRNAs produced by MenT3. Indeed, additional experiments demonstrated that RNase PH did not counteract MenT3 in the cell. However, further experiments using in vitro assay and in vivo complementary analysis uncovered that CCA-adding enzyme (PcnA) regenerates intact tRNAs in vitro and in the cell. Furthermore, the authors uncovered the substrate recognition mechanism of MenT3 through in-depth biochemical experiments. This work revealed a novel mechanism by which the intact tRNA 3'end is controlled by toxin and tRNA repair enzyme. This reviewer now highly recommend to publish this manuscript in Nature Communications.

Reviewer #3 (Remarks to the Author):

The authors have performed additional experiments and also have provided clarifications to my query. Overall, I am satisfied with the revised manuscript.